

# Investigation of observational error sources in multi Doppler radar vertical air motion retrievals: Impacts and possible solutions

Mariko Oue[1], Pavlos Kollias[1,2,3], Alan Shapiro[4], Aleksandra Tatarevic[3], Toshihisa Matsui[5]

[1]School of Marine and Atmospheric Sciences, Stony Brook University, Stony Brook, 11794, USA
[2]Environmental and Climate Sciences Department, Brookhaven National Laboratory, Upton, 11973, USA
[3]Department of Atmospheric and Oceanic Sciences, McGill University, Montreal, H3A 0G4, Canada
[4]School of Meteorology, University of Oklahoma, Norman, 73019, USA
[5] Mesoscale Atmospheric Processes Laboratory NASA Goddard Space Flight Center, Greenbelt, 20771, USA

*Correspondence to*: Mariko Oue (mariko.oue@stonybrook.edu)

**Abstract.** Multi-Doppler radar network observations have been used in different configurations over the last several decades to conduct three-dimensional wind retrievals in mesoscale convective systems. Here, the impact of the selected radar volume coverage pattern (VCP), the sampling time for the VCP, the number of radars used, and the added value of advection correction on the retrieval of the vertical air

motion in the upper part of convective clouds is examined using the Weather Research and Forecasting (WRF) model simulation, the Cloud Resolving Model Radar SIMulator (CR-SIM) and a three-dimensional variational multi-Doppler radar retrieval technique. Comparisons between the model truth (i.e., WRF kinematic fields) and updraft properties (updraft fraction, updraft magnitude, and mass flux) retrieved from the CR-SIM-generated multi-Doppler radar field are used to investigate these impacts. In

overall, the VCP elevation strategy and sampling time is found to have a significant effect on the retrieved updraft properties above 6 km altitude. Retrievals conducted using a 2-min or shorter VCPs show small impacts on the updraft retrievals, and the errors are comparable to retrievals using a snapshot cloud field. Increasing the density of elevations angles and/or an addition of data from one more radar can reduce this uncertainty.  It is found that the VCP with dense elevation angles appears to be more effective than the

addition of data from the fourth radar, if the VCP is performed in 2 minutes. The use of dense elevation angles combined with an advection correction applied to the 2-min VCPs can effectively improve the updraft retrievals. For longer VCP sampling periods (5 min) the errors are considerably larger, and the value of advection correction is challenging due to the rapid deformation of the dynamical structures in the simulated mesoscale convective system. This study highlights several limiting factors in the retrieval





of upper-level vertical velocity from multi-Doppler radar networks and suggests that the use of rapid-scan radars can substantially improve the quality of wind retrievals if conducted in a limited spatial domain.

# 1 Introduction

Measurements of vertical air motion in deep convective clouds are critical for our understanding of the dynamics and microphysics of convective clouds (e.g., Jorgensen and LeMone, 1989). Convective mass flux is responsible for the transport of energy, mass and aerosols in the troposphere, which significantly impact large-scale atmospheric circulation and local environment and affect the probability of subsequent formation of clouds (e.g., Hartmann et al., 1984; Su et al., 2014; Sherwood et al., 2014). Consequently, the vertical air motion estimates are widely employed to improve convective parameterizations in global model (e.g., Donner et al., 2001) and also to evaluate the cloud resolving model (CRM) simulations and large eddy simulations (LES, e.g. Varble et al., 2014; Fan et al., 2017).

Aircraft penetration of convective clouds offer the most direct method to measure the vertical air motions (e.g. Lenschow, 1976), however, practical hazards and operational costs have resulted in a valuable but limited dataset (e.g., Byers and Braham, 1948; LeMone and Zipser, 1980). Current aviation regulation does not permit such penetration anymore. Ground-based and airborne profiling Doppler radars provide a high degree of detail of convective clouds in both time and height and can sample even the most intense convective cores (e.g., Wakasugi et al., 1986; Heymsfield et al., 2010; Williams, 2012; Giangrande et al., 2013; Kumar et al., 2015). One drawback of profiling radar techniques is their limited sampling of individual storms and the lack of information on the temporal evolution of the convective dynamics and structure, thus, making their use in model evaluation challenging.

Since the pioneering work of Lhermitte and Miller (1970), networks of two or more scanning Doppler radars and the use of multi-Doppler radar wind retrieval techniques have been widely used to overcome aforementioned limitations (Junyent et al., 2010; North et al., 2017). In addition to research radars, operational Doppler radar networks can, in certain conditions, accomplish a large coverage of multi-Doppler radar retrievals (e.g., Bousquet et al., 2007; Dolan and Rutledge, 2007, Park and Lee, 2009). While various Doppler radar wind retrieval techniques have been proposed (Chong and Testud, 1996; Chong and Campos, 1996; Bousquet and Chong, 1998; Gao et al., 1999; Protat and Zawadzki, 1999),





three-dimensional variational (3DVAR) techniques are commonly used because of its robust and reliable solutions by minimizing errors (e.g., Potvin et al., 2012b).

Multi-Doppler radar analysis have been used to better understand mesoscale dynamics, low-level divergence, and microphysical-dynamical interactions (e.g., Kingsmill and House, 1999; Friedrich and Hagen, 2004; Stonitsch and Markowski, 2007; Collis et al., 2013; Oue et al., 2013, and many others). There is also considerable literature discussing different sources of uncertainties in dual- or multi-Doppler radar wind retrieval. The interpolation and smoothing techniques used (Cressman, 1959; Barnes, 1964, Given and Ray, 1994) can have an impact on the quality of Doppler radar wind retrieval (e.g., Collis et al., 2010). Another source of uncertainties is related to the hydrometeor fall speed estimates (e.g., Steiner, 1991; Caya, 2001) especially at shorter wavelengths (e.g., X and C bands) where the signal attenuation can bias the estimates. Clark et al. (1980) focused on a difference in spatial scales of convective cloud systems and an impact of radar sampling volume smaller than model spatial resolution. Bousquet et al. (2008) estimated uncertainties in wind fields from their operational multi-Doppler radar retrieval by simulating radar measurements using numerical model output. They pointed out that missing low-level measurements and poor vertical sampling could produce significant uncertainties in retrieval of low-level wind fields. These investigations are conducted by formulating suitable Observing System Simulation Experiments (OSSEs). Potvin et al. (2012b) investigated potential sources of errors in multi-Doppler radar wind retrievals for supercell observations using OSSEs. They suggested that the magnitudes of vorticity and its tendency fields were sensitive to the smoothness constraint in the analysis, and assumptions of spatially constant storm motion and no storm-evolution led to significant errors in middle and upper levels.

A common result from the studies above is that the uncertainties increase with height because scanning radar data density inevitably becomes lower at higher altitudes. Meanwhile, deep convective clouds generally show maximum updrafts at middle and upper parts of the clouds (e.g., Giangrande et al., 2013). Here, we are concerned with the retrieval uncertainties of vertical air motion especially in the middle and upper levels of deep convective clouds. The motivation for this study is two folded. First, the US Department of Energy (DOE) Atmospheric Radiation Measurement (ARM) program operates an atmospheric observatory at Southern Great Plains (SGP), Oklahoma (Mather and Voyles, 2013), where





scanning Doppler radars and profiling instruments provide unique dynamical and microphysical measurements. During the Midlatitude Continental Convective Clouds Experiment (MC3E, Jensen et al., 2016), the ARM precipitation scanning Doppler radars accomplished dense network of Doppler radar measurements of deep convective clouds explicitly designed to retrieve three-dimensional (3D) wind

(North et al., 2017). However, our experience with the data and a series of experiments performed in this study suggest that despite the plethora of radar systems at the ARM SGP observatory, the 3D wind retrievals are subject to large errors especially at the upper levels. It is possible that some of the errors are associated with radar volume coverage pattern strategy that does not satisfy the requirement for high spatiotemporal observations, which have been highlighted in recent studies with high-resolution CRM

simulations of convective cloud properties (e.g., Morrison et al., 2015; Hernández-Deckers and Sherwood, 2016). Second, the paucity of available datasets of vertical air motion limits our ability to quantitatively analyze structures and characteristics of the mesoscale convective systems (MCSs) and evaluate model outputs of the MCSs (e.g., Varble et al., 2014; Liu et al., 2015; Donner et al., 2016; Fan et al., 2017). Thus, we are interested in determining the sampling capabilities required for a multi-Doppler

radar network to address these errors and investigating if radar networks based on different technology (e.g., phased-array radars, Otsuka et al., 2016; Kollias et al., 2018a) can address these errors.  To do so we focus on impact of the multi-Doppler radar network setup and not how we quality-control, interpolate or use the Doppler radar observations in a minimization routine. The latter is the same in all the experiments performed here and is described in North et al. (2017). We are investigating the impact of

the selected radar volume coverage pattern (VCP), the sampling time for the VCP, the number of radars used and the added value of advection correction upon the uncertainties of multi-Doppler radar wind retrieval.

## 2 Data and methodology

OSSE studies are generally used to assess impacts of operational observing systems on, for example,

observation-based value-added products and weather forecasts (Timmermans et al., 2009). The OSSE conducted in this study is composed of the following steps:



1) Produce the set of simulation data by a high resolution numerical weather model of a convective cloud system and generate the model hydrometeor and dynamical fields at a high temporal resolution to capture the storm evolution at scales unresolved by typical VCP's;

2) Use a sophisticated radar simulator to reproduce the VCP of a multi-Doppler radar system and produce radar observables at radar coordinates with the realistic radar characteristics (beamwidth, range resolution and sensitivity);

3) Grid the simulated radar observations to a Cartesian coordinate and conduct a variational 3D multi-Doppler wind retrieval algorithm to estimate the dynamical field; and

4) Evaluate the retrieved wind field against the corresponding field from the numerical model direct output.

The Weather Research Forecasting model is used to produce simulation of an MCS case on 20 May 2011 observed in Oklahoma during the MC3E (step 1). The WRF output is used as an input to the Cloud Resolving Model Radar SIMulator (CR-SIM; Tatarevic et al., 2018) to simulate radar reflectivity and Doppler velocity from scanning radars (step 2). The simulated radar reflectivity and Doppler velocity fields are then resampled and converted into radar polar coordinate according to VCPs (step 2). The radar reflectivity and Doppler velocity fields at radar polar coordinate are converted into the Cartesian grid, and then they are used to estimate 3D wind field using the 3DVAR multi-Doppler radar wind retrieval algorithm developed by North et al. (2017) (step 3). The retrieved vertical velocity fields are compared against the WRF-simulated dynamical field to investigate impacts of the limitations attributed to the radar observations and retrieval technique on the retrieved vertical wind field (step 4).

## 2.1    WRF Simulation for 20 May 2011 MCS

The WRF simulation horizontal domain is 960 km x 720 km with 0.5 km horizontal grid spacing. The vertical resolution varies from approximately 30 m near the surface to 260 m at 2 km altitude and maintains this resolution approximately constant above 2 km altitude. To include time evolution in volume scan coverage pattern, the WRF simulation provides output at every 20 seconds. The Morrison double moment microphysics scheme was used, which predicts mass and number mixing ratios for liquid cloud, rain, ice cloud, snow, and a medium density lump graupel representing the rimed ice with a switch





to modify the settings for graupel to a high density hail (Morrison et al., 2005). Tao et al. (2016) pointed out that simulations including the hail option better represented the observed MCSs during the MC3E period than those not using hail. In their study for the May 20 MC3E case, Fridlind et al. (2017) in their study used the Morrison double moment microphysics scheme with the hail option. The present study

also applies the hail category to the simulation instead of graupel. This case has been actively analyzed for its dynamical and microphysical structures (e.g., Liu et al., 2015; Wu and McFarquhar, 2016; Fan et al., 2017). In this study, we treat the WRF-simulated vertical velocity field as "truth" to evaluate the performance of multi-Doppler radar wind retrieval.

## 2.2    CR-SIM Simulation of 20 May 2011 MCS case

The CR-SIM is a sophisticated radar forward operator developed to bridge the gap between high-resolution cloud model output and radar observations (Tatarevic et al., 2018). The CR-SIM can be applied on the 3D model output produced by a variety of CRM and LES, such as WRF, Regional Atmospheric Modelling System (RAMS), System for Atmospheric Modelling (SAM), and the ICOsahedral Nonhydrostatic (ICON) model. It emulates the interaction between transmitted polarized radar waves and

rotationally symmetric hydrometeors and can simulate the power (radar reflectivity), phase (Doppler velocity) and polarimetric (specific differential phase, differential reflectivity, depolarization) variables with a fixed elevation angle or varying elevation angles with respect to a specified radar location.

Several experiments are performed to evaluate the limitations of the sensing techniques employed in the network of three X-band Scanning ARM Precipitation Radars (X-SAPRs, named I4, I5, and I6,

respectively) at the SGP site (Fig. 1), which provided high-resolution radar observations of convective systems during the MC3E (e.g., North et al 2017). The ARM SGP network is selected because it is comprised by three identical radar systems that are employed together and can be operated in a coordinated manner. Furthermore, since it is a long-term facility for the study of deep convective clouds, it is important to assess the capability and uncertainties. Using CR-SIM, we simulated measurements of

the three X-SAPRs. In order to investigate the impact of an increased number of radars, observations from the C-band Scanning ARM Precipitation Radar (C-SAPR) at the SGP site (Fig. 1) are also simulated. Characteristics and settings of the simulated radar measurements are shown in Table 1. To investigate the





impact of increasing the number of elevation angles and the maximum elevation angle, a VCP including additional elevation scans for the X-SAPR measurements is introduced. These simulations with X-SAPR aim to examine effects of using faster scanning radars such as the Doppler on Wheels (DOW, Wurman, 2001), the Atmospheric Imaging Radar (AIR, Isom et al., 2013), the Rapid scanning X-band polarimetric

(RaXPol, Pazmany et al., 2013) and low-power X-band phased array radars (LPAR, Kollias et al., 2018a). Locations of radars used in this study and the simulated retrieval domain are shown in Fig. 1. Details about the elevation angle settings are described in Sect. 2.4.

The retrieval simulation domain size is 50 km × 50 km × 10 km above the ground level (AGL) centered around the ARM SGP Central Facility (CF). In the simulations, CF and the domain were virtually located

within a vigorous convective region of the MCS to capture the intense vertical velocity (Fig 1b). We assume that the lowest boundary of the simulation domain is idealized as flat at the ground level of 0.3 km above sea level.

For each radar, the CR-SIM forward simulated reflectivity and Doppler velocity are provided at the WRF grid coordinate by CR-SIM. They are then converted into radar polar coordinates considering all

the radar characteristics that control the spatial resolution of radar observations (range weighting function, antenna beamwidth, and VCP strategy). The settings shown in Table 1 are consistent with the settings used during the MC3E period. For each radar the minimum detectable signal ($Z_{min}$) curve, which is attributed to the number of samples integrated for each radar sampling volume, is estimated using an equation $Z_{min}(r) = C + 20\log_{10}(r)$. In this equation, $Z_{min}$ is expressed in logarithmic units (dBZ) with

the range $r$ (distance from the radar) in km, and the constant $C$ that depends on the radar system characteristics expressed in dBZ; $C$ = -40 for X-band radars and $C$ = -35 for C-band radar are used in this study. These values are similar to those for X-SAPRs and C-SAPR at the SGP site.

## 2.3   Wind Retrieval

The 3DVAR wind retrieval technique described in North et al. (2017) is used to estimate the 3D wind

field. The wind retrieval algorithm inputs the Cartesian coordinate reflectivity and Doppler velocity fields from each radar and uses 3DVAR technique continuity constraint proposed by Potvin et al. (2012a), which capitalizes on the physical constraints of radar radial velocity observations, anelastic mass continuity,





surface impermeability, background wind field, and spatial smoothness. Details of the constraints are described in North et al. (2017).

The simulated radar reflectivity and Doppler velocity with the radar polar coordinate are converted to the Cartesian coordinates for each radar measurement at horizontal and vertical spacings of 0.25 km using

a single-pass isotropic Barnes distance-dependent weight (Barnes, 1964) with a constant smoothing parameter $\kappa$.

$$w_{i,q}(d) = \exp\left(\frac{-d^2}{\kappa}\right) \forall\, i = 1, \ldots, n \text{ and } q = 1, \ldots, Q \tag{1}$$

Here $w_{i,q}$ is the weight for grid box $i$ and radar gate $q$ separated by distance $d$. At each grid box radar moments are estimated using the nearest 200 radar data gates with weights (Eq. 1) using $\kappa = 0.13$ km$^2$ for

interpolation. The cutoff distance is determined as the distance where the weight is less than 0.01 (d $\approx$ 0.8 km). These parameters are chosen so that the statistical error in retrieved vertical velocity is minimal for the present case. Generally, data density at constant altitudes decreases with height and when increasing a distance from radar. Figures 2c-f show distance to the nearest radar data point at each Cartesian grid box at constant altitudes. These settings for gridding are fixed for all radar simulations, and this study

does not consider uncertainties attributed to the settings for gridding process. The gridding technique has been well optimized in North et al. (2017), and the uncertainties in the gridding method and data smoothing processes have been well investigated in previous studies (e.g. Majcen et al., 2008; Potvin et al., 2012a).

There are several important sources of errors when considering the retrieval of vertical motion in

convective systems other than the radar VCP, the most important among them are: not unfolded correctly observed Doppler velocity, estimation of hydrometeor fall velocities, attenuation correction, and assumption of background environments. In all experiments in this study, Doppler velocity folding is disabled as an option, thus, the radial Doppler velocities are unfolded correctly. This eliminates the possibility of errors being introduced by incorrect Doppler velocity unfolding.

The difference between the "true" hydrometeor fall velocity $V_f$ and the assumption based on an empirical formula that relates $V_f$ with the radar reflectivity (e.g., Caya, 2001) can be a possible source of errors in wind retrievals (e.g., Potvin et al., 2012b; North et al., 2017). In the WRF simulations used here, $V_f$ is parameterized depending on the microphysics scheme as a function of particle diameter. The



hydrometeor's fall speeds ($V_f$) are given as a function of the hydrometeor diameter ($D$) and altitude ($h$) in a form:

$$V_f(h, D) = f_c(h) \cdot a_v \cdot D^{b_v} \tag{2}$$

where $a_v$ and $b_v$ are coefficients, and $f_c(h) = \left(\rho_{surf}/\rho(h)\right)^{k}$ is the correction factor for air density ($\rho(h)$:

air density at height $h$, $\rho_{surf}$: surface air density) with exponent $k$ (Morrison et al., 2005; Tatarevic et al., 2018). In the CR-SIM, reflectivity-weighed mean velocity is computed at each grid box in the following manner. The hydrometeor fall speeds as a function of the hydrometeor diameter are averaged over the diameter range with weights that are proportional to the CR-SIM estimated reflectivity for each hydrometeor particle size, and then the mean hydrometeor fall speeds are again averaged over all

hydrometeor types present in each grid box with weights of reflectivity. In all experiments in this study, the simulated reflectivity-weighted mean $V_f$ are used in the retrieval, thus, no error attributed to the fall velocity estimates is introduced in the wind retrieval technique.

Another source of errors is the impact of signal attenuation by the hydrometeors along the propagation path, especially at C-band and X-band radar measurements. Since the attenuation is unknown, any

attenuation-corrected radar reflectivity acts as a possible error source in the wind retrievals, particularly for hydrometeor fall speed estimates. However, as previously specified, the hydrometeor particle size distributions and $V_f$ used in this study are the ones prescribed by the WRF model microphysics, thus, no error is introduced.

Finally, background horizontal wind vector, temperature, and air density are obtained by averaging

WRF output values over the retrieval domain at each altitude and are used in place of sounding measurements over the SGP CF site. Although this study does not consider uncertainties in the background assumption, the change in the background data would have small impact on the retrieved updraft velocities as discussed in North et al. (2017).

## 2.4    Settings for wind retrieval experiments

Three factors influencing the updraft velocity estimates are investigated. The first is radar volume coverage pattern (VCP) which determines the set of elevation angles used by the radars to sample the



volume of the analysis domain. The second is time interval needed by the radars of the network to complete the specified VCP to emulate both the advection and temporal evolution of the convective cloud system. Third, the added value of the advection correction for the different sets of VCP settings is evaluated. The experiments and their names are listed in Table 2.

### 2.4.1 Control wind retrieval simulation (3FullGrid)

The control wind retrieval simulation is an ideal, instantaneous VCP where all radars of the network sample all the WRF grid points. As a result, three measurements of radar reflectivity and radial Doppler velocity from the three X-SAPRs are available at each grid box of the WRF grid (named 3FullGrid). This experiment does not undergo the conversion process from the WRF grid to radar coordinate or the gridding process from radar coordinate to the Cartesian coordinates. Therefore, this does not include uncertainties from VCP or radar characteristics (beamwidth and range-bin spacing). Thus, the retrieved wind field should be a very good estimate of the true wind field and only the potential uncertainty in the wind retrieval algorithm can affect its quality. In this OSSE, the 3FullGrid is used for an upper bound of the performance of any of the conducted experiments and also serves as a sanity check for the wind retrieval algorithm.

### 2.4.2 Radar VCP

In a typical radar VCP, the number of elevation angles depends on the antenna scan rate and the desired time period for completing the VCP (typically 5-6 min). The antenna scan rate depends on the pedestal technical specifications and the minimum number of radar samples needed to estimate the radar observables with low uncertainty. The elevation angles are generally tightly selected at low elevations to provide good coverage over long horizontal distances and relatively sparse at higher elevations as the X-SAPR's VCP shown in Table 1 and Figure 1c.

In the experiments performed here the impact of an increased number of elevations angles especially at high elevations is investigated while the antenna beamwidth, range-gate spacing, and maximum unambiguous range are kept unchanged and similar to the radar settings during MC3E. The following VCP are used: i) three X-SAPRs with the general VCP which is the same as during MC3E (named 3XR,



Fig. 1c); ii) three X-SAPRs with denser elevation angles (named 3LR, Fig. 1d; the name "LR" stands for low-power X-band phased array radar, LPAR, Kollias et al., 2018a); and iii) same as i) but the C-SAPR measurements are added (named 4SR). Details of the VCPs are shown in Table 1. The settings i) and iii) use general VCPs for X-SAPR and C-SAPR which are the same as those during MC3E. The X-SAPR VCP is composed of 21 elevation angles ranging from 0.5° to 45°, and the C-SAPR VCP is 17 elevation angles ranging from 0.75° to 42°. Elevation angles for the setting ii) are equally distributed from 0.5° to 59.5° with a 1° increment; in total there are 60 elevation angles. This elevation setting intends to simulate rapid scanning radar observations.

The selection of the VCP (XR or LR) affects the density (spacing) and availability of observations at each height for gridding. Figures 2a and 2b show the coverage from the three radars for the retrieval domain for 3XR and 3LR VCPs, respectively. The cone of silence (absence of radar observations) from each radar is represented as yellow circle, in the middle of which the X-SAPR is located. Within the cone of silence of each radar, we only have two available radar measurements for the wind retrieval. In addition to the availability of radar observations, the spacing of the radar observations affect the quality of the gridding. Regions including few radar data points, particularly higher elevation angle regions for the XR VCP, may need to interpolate radar data at longer distances from the grid points. Figures 2c-2f show distance of the nearest radar data point at each grid box at heights of 1 km and 8 km for X-SAPR I6, and Figs. 2g and 2h show normalized histograms of the nearest distance. At lower altitudes, the nearest distances in the entire retrieval domain (thin lines in Fig. 2h) are mostly less than 0.3 km for both VCPs. At higher altitudes (thin lines Fig. 2h), the distances of the nearest radar data points from the LR VCP are same as at lower altitudes, indicating that the LR VCP has similar radar data density at higher and lower altitudes. For the XR VCP, in contrast, many of grid boxes at 8 km AGL needed to use radar data at distances farther than 0.4 km, resulting in stronger smoothing when the gridding process.

### 2.4.3 Time duration of the radar VCP

Three time periods are considered here for the completion of the radar network VCP: i) snapshot (named Snap), where it is effectively assumed that the first WRF model output (at time 0 sec, top row, Fig. 3) is frozen in time and the radars instantaneously collect data according as their VCP without any





cloud evolution; ii) a 2 minute (named 2min) radar network VCP to emulate the performance of rapid scanning radar networks; and iii) a 5 minute (named 5min) radar network VCP to emulate the performance of the ARM SGP network during MC3E and the performance of other mechanically-scanning radar networks. The 3FullGrid simulation (Sect. 2.4.1) uses a Snap VCP. The Snap VCP eliminates any
concerns regarding advection and temporal evolution of the convective cloud and is used as benchmark of performance.

A set of WRF simulations at different times is used to construct the Plan Position Indicator (PPI) scans of the VCP; if a PPI scan takes more than 20 seconds, the WRF output in the following time step is used for the next PPI scan. An example demonstrating how different WRF model outputs are used in this
experiment is shown in Fig. 3. Figure 3 shows horizontal cross sections of the radar reflectivity and vertical velocity at 7 km and a vertical cross section of the at the area indicated with the solid line in the horizontal cross sections. The snapshot simulations use the WRF model output data at 12:18:00 UTC (top row). The 2-min VCP simulations use the WRF model output data from six consecutive model outputs extracted from 12:18:00 UTC to 12:19:40 UTC every 20 seconds. Each model output is used to forward
simulate 3-4 PPI scans from the C-SAPR and the X-SAPR's when nominal (MC3E) VCP elevation angles are used (3XR and 4SR) and 10 PPI scans for X-SAPR simulations when the denser elevations angles VCP is simulated (3LR). The corresponding plots for the latest model output (12:19:40 UTC) used to forward simulate the highest elevations of the 2-min VCP are shown in Fig. 3 (middle row). In accordance, the 5-min VCP simulations use the WRF data for 5 minutes composed of 15 snapshots
ranging from 12:18:00 UTC to 12:22:40 UTC every 20 seconds. Each snapshot data was used for 1-2 PPI scans for C-SAPR and X-SAPR simulations with general VCP elevation angles (3XR and 4SR) and 4 PPI scans for denser VCP elevation angles (3LR). The corresponding plots for the latest model output (12:22:40 UTC) used to forward simulate the highest elevations of the 5-min VCP simulations is shown in Fig. 3 (bottom row).

**2.4.4   Advection correction**

The high temporal resolution WRF output allows us to evaluate the impact of advection and evolution of the cloud field during the time period needed to complete the radar network VCP. If the cloud field





was frozen (no cloud evolution), horizontal advection is expected to tilt the cloud and dynamical structures in vertical as the cloud system moved in a certain direction. Advection schemes have been proposed to address this issue (e.g. Protat and Zawadzki, 1999; Shapiro et al., 2010b; Qiu et al., 2013). The present study used a reflectivity-based spatially-variable advection correction scheme described in

Shapiro et al. (2010a) which allows trajectory of individual clouds and smooth grid-box-by-grid-box corrections of cloud locations. This scheme takes into account changes in cloud shape with time by using two different time PPI scans. The advection correction process is similarly implemented in this case.

The advection correction is applied between two similar elevation angle PPIs from consecutive VCPs. Each simulated radar reflectivity field in PPI is converted and projected onto the two-dimensional (2D)

Cartesian coordinate plane at a spatial resolution of 250 m. We used a smoothness weighting coefficient of 300 dBZ$^2$ in a cost function in the technique. Using two 2D Cartesian coordinated PPI data at two different times at the same elevation angle, the advection correction algorithm performs horizontal trajectory analysis of reflectivity and estimates the reflectivity pattern translation components U and V on the 2D surfaces for each VCP elevation angle. The pattern translation components U and V fields

along with the associated trajectories of virtual particles moving with the reflectivity field are then used to effect the advection correction of the radial wind field according to a time difference between a PPI scan and the base PPI scan, when creating the 3D Cartesian coordinated data. Such processed simulated radar measurements in 3D Cartesian coordinates are then incorporated into to the 3DVAR algorithm for the 3D wind retrieval as described in Sect. 2.3.

However, the cloud and dynamical field evolve while advected. This results in observing different cloud life stages by different PPI scans. Figure 3 (right column) shows a vertical cross-section of the vertical air motion within a convective cell that is tracked using the WRF model output. The location of the convective cell and vertical distributions of updrafts and downdrafts significantly vary from 12:18:00 UTC to 12:22:40 UTC. Thus, we need to consider that gridded radar observations collected after the

completion of the VCP do not represent an actual snapshot of the 3D convective dynamics. Consequently, the mass continuity constraint will be applied in the column of gridded radar observations that is a mosaic of different stages of the lifetime of a convective element, and this, in turn, will limit the ability for this 3DVAR approach to satisfy the mass continuity equation (e.g., Clark et al., 1980; Gal-Chen, 1982),





resulting in large uncertainties of the wind retrievals. The experiments presented here are designed to quantify the impact of cloud evolution on the retrieved wind field (Sect. 3.4).

# 3 Results

The evaluation of multi-Doppler radar-based velocity retrievals using independent observations is
challenging to perform (e.g., Collis et al., 2013; North et al., 2017). Profiles of percentiles of updraft magnitudes are often used to evaluate numerical model results against vertical velocity retrievals from scanning Doppler radar networks and/or profiling radars (e.g., Wu et al., 2009; Varble et al., 2014; Fan et al., 2018). Here, we are interested in the estimation of the convective mass flux, thus, profiles of updraft morphology (number and area) and intensity (magnitude) are used to represent the impact of the selected
sampling strategy.

## 3.1    Evaluation of multi-Doppler radar updraft property retrievals

Horizontal cross sections at 7 km AGL and vertical cross sections along y = 0 km of the retrieved vertical velocity field from the XSAPR network using the original grid (3FullGrid) and using the standard (XR) VCP for three different time periods (Snap, 2min and 5min) are shown in Fig. 4 (b, c, d, and e,
respectively). The WRF model out at t = 0 (12:18:00 UTC) is also shown in Fig. 4a.  The selection of the height of 7 km is based on the WRF model output analysis: the chosen height is the one with maximum updraft values.  The WRF model output vertical velocity field indicates the presence of several cell-like, horizontally coherent updraft structures with updraft magnitude exceeding 5 m s$^{-1}$. The 3FullGrid simulation (Fig. 4b) provides results in good agreement with the original WRF vertical velocity field (Fig.
4a), suggesting that the 3DVAR wind retrieval algorithm is performed well.

The snapshot simulation (3XRSnap, Fig. 4c) provides results that are comparable to the original WRF vertical velocity field and 3FullGrid retrieved vertical velocity field at 7 km AGL, but slightly overestimates the updraft velocity above 8 km AGL (Figs. 4a and 4b). The 3XRSnap simulation reproduces the location and size of the stronger updraft areas defined with updraft magnitudes above 5 m
s$^{-1}$, which show the cell-like structures, but it tends to have higher uncertainty in the areas around the location of strong convection (vertical velocity < 5 m s$^{-1}$). The uncertainty is attributed to the selected





radar VCP, rather than the 3DVAR wind retrieval algorithm. As increasing VCP time periods (2 min and 5 min) shown in Figs. 4c and 4d, respectively, the retrieved velocity features became less sharp, broader and shifted in space. The retrieved vertical velocity field shows the impact of gridding sparse observations (ring structures representing the poor X-SAPR sampling at 7 km) and the vertical velocity features appear

elongated and connected.

At any vertical level in the WRF model output and in the retrieved 3D velocity field, a convective updraft core is defined as an area larger than 0.5 km$^2$ and with updraft velocities higher than 5 m s$^{-1}$. Figure 5a displays the profiles of the number of updraft cores from the 3FullGrid control wind retrieval simulation and from the WRF snapshot data at 12:18:00 UTC, WRF 2-min average (12:18:00-12:19:40),

and WRF 5-min average (12:18:00-12:22:40). As expected, the 3FullGrid retrieved profile of number of updraft cores captures very well the profile of the number of updraft cores in the WRF snapshot model output. Differences appear small between WRF Snap and 3FullGrid and are attributed to the potential uncertainty in the retrieval algorithm. The 2- and 5-min WRF output averaged profiles suggest that the number of convective updraft cores does not change over a period of 2 to 5 min. Figures 5b-e demonstrate

performance of the 3DVAR wind retrieval for several different configurations as described in Table 2. Noticeable departure between the WRF direct model output (number of updraft cores) and the estimated number of updraft cores above 6 km AGL is observed for all the detecting configurations with the exemption of the LR VCP. The use of a fourth radar or the implementation of the advection correction has little to no impact on the findings. The retrieved profiles of the number of coherent updrafts structures

show little sensitivity to the VCP time. This can be attributed to the fact that the number of updraft coherent structures does not change within the 5 min required to complete all sampling strategies. Another possibility is that any stretching/distortion of the coherent structures due to cloud evolution and advection does not results to changes in the number of coherent structures.

In a similar manner, the retrieved updraft fraction (UF, Fig. 6), the retrieved convective mass flux (MF,

Fig. 7) and the mean updraft velocity ($\overline{w}$, Fig. 8) for the different VCPs are investigated and compared to the direct model output. In this study, convective mass flux (MF) is estimated at each height as:

$$MF = UF \ \overline{w} \ \overline{\rho_d} \quad [kg \ s^{-1} \ m^{-2}] \tag{3}$$





where UF is updraft fraction over the domain, $\overline{w}$ is mean vertical velocity over the updraft area, and $\overline{\rho_d}$ is dry air density averaged over the domain. The updraft fraction and mean updraft velocity strongly impact the domain averaged convective mass flux, which can be used to understand mass, energy and aerosol transport by the convective system.

5    A comparison limited to a smaller domain where the higher density radar observations are available (squared area in Fig. 2) is added (Figs. 6-8, panel f). Furthermore, the analysis is presented for the two different updraft thresholds: 5 m s$^{-1}$ (UF$_5$) and 10 m s$^{-1}$ (UF$_{10}$). In contrast to the number of coherent updraft cores, the profiles of UF, MF and $\overline{w}$ exhibit larger sensitivity to the sampling parameters. Here, the results are described and a more detail analysis of the impact of the different options in the observational setup are discussed in subsequent sections.

10    Figure 6 displays updraft fraction (UF) profiles from the different simulations for the two above mentioned updraft thresholds UF$_5$ and UF$_{10}$. Each panel shows UF from the WRF snapshot at 12:18:00 by a black solid line for the comparison. Figure 6a also compares the WRF snapshot at 12:18:00, the WRF 2-min average, and the WRF 5-min average. The UF profiles from both WRF 2-min average and the WRF 5-min average are in very good agreement with that from WRF snapshot; this consistency is also shown in MF and $\overline{w}$ profiles (Figs. 7a and 7b, respectively), indicating that the updraft properties are statistically similar throughout the 5 minutes in this case. As expected, the profiles from 3FullGrid simulation are in very good agreement with the WRF output for all thresholds (Fig. 6a) but show an underestimation by ~0.01 at 5.3 km AGL. For reference, 1% difference in the updraft fraction corresponds to 25 km$^2$ for a 50 km × 50 km retrieval domain. All the retrieved profiles of coherent updraft fraction exhibit considerable differences with the WRF output above 6 km AGL (Fig. 6b, d, e). In general, the retrieved updraft fractions increase above 6 km AGL while the WRF output indicates that the updraft fraction decreases.

Figures 7 and 8 show MF and $\overline{w}$ profiles, respectively, from simulated wind retrievals together with those from the WRF output. The MF and $\overline{w}$ profiles in the Figures 7 and 8 are coupled with updraft areas for velocities larger than 10 m s$^{-1}$ (MF$_{10}$, $\overline{w}_{10}$) and for velocities lather than 5 m s$^{-1}$ (MF$_5$, $\overline{w}_5$). For the WRF output, the peaks of MF values are found at heights between 5 and 7 km AGL, and the MF$_{10}$ values are generally the half of MF$_5$. The 3FullGrid simulation (Fig. 7a) well captures those features, but the



maximum values at 5.25 km AGL are slightly underestimated as $MF_5$ decreases by up to 0.05 kg s$^{-1}$ m$^{-2}$. Since the $\overline{w}$ values are well estimated, the underestimation is driven by the small underestimation of UF (by 0.01, Fig. 6a).

The mean updraft velocities for both $UF_{10}$ ($\overline{w}_{10}$) and $UF_5$ ($\overline{w}_5$) from 3LRSnap slightly increase above 6 km AGL (Fig. 8c). Consequently, the $MF_5$ profile is improved as it increases at 4.5-7 km and decreases above 7 km (Fig. 7c). Similarly, the $MF_{10}$ profile is also improved as it increases above 4.5 km, but it still underestimated by 0.05 at 5-9 km AGL. Compared to the same VCP periods, the 3LR retrievals also show similar improvements at 2-min VCP and 5-min VCP. These results suggest that the VCP with dense elevation angles can improve the retrieval of strong updrafts with velocities larger than 10 m s$^{-1}$, and is more effective at higher altitudes (> 8 km).

## 3.2    Effects of VCP elevation sampling and number of radars

The impact of the maximum elevation angle and density of elevation angles used in the VCP is easily demonstrated when comparing the 3XRSnap and 3LRSnap retrievals for the entire domain or within the smaller domain (square area in Fig. 2). For all updraft parameters investigated here (number of updraft cores, UF, MF, and $\overline{w}$), the 3LRSnap produces improved comparisons to the direct model output especially when limiting the evaluation area to the center square domain. The comparison for the number of cloud cores (Fig. 5) shows that 3XRSnap overestimated above 6.5 km. Figure 6b shows that $UF_5$ values from the 3XRSnap are overestimated above 6.5 km AGL, while $UF_{10}$ values above 6.5 km are underestimated. These profiles indicate that updraft areas of 5-10 m s$^{-1}$ are overestimated for the 3XRSnap retrievals. Thus, the overestimation of the number of updraft cores is caused by overestimation of updraft areas of 5-10 m s$^{-1}$. This feature is also shown in other snapshots and 2-min VCP retrievals. The impact of a longer time VCP is more pronounced in the UF retrievals than the number of coherent updrafts cores. As in the case for the profile of the number of coherent updraft cores, the use of the LR VCP improves the updraft fraction profile retrievals.

The $UF_{10}$ values from the 3XRSnap simulation are underestimated by 0.01 at 5-7 km AGL (~30 % of the true fraction, Fig. 6b) at higher altitudes above 5 km. The errors generally increase with height above 6 km AGL. This result is similar to the dual-Doppler radar wind retrieval OSSE study for supercell storms





by Potvin et al. (2012b). The mean updraft velocities are also underestimated by 1 m s$^{-1}$ for UF$_{10}$ above 5.5 km. The underestimations in $\overline{w}_{10}$ and UF$_{10}$ profiles result in underestimation of MF$_{10}$, and the maximum underestimation of 0.1 kg s$^{-1}$ m$^{-2}$ is found at 6 km AGL. For the threshold of 5 m s$^{-1}$, the overestimation of UF$_5$ above 7 km results in overestimation of MF$_5$, while the underestimation of the

mean updraft velocity by 2 m s$^{-1}$ above 4.5 km for UF$_5$ leads to the underestimation of MF$_5$ at 4.5-7 km AGL.

      Substantially improved retrievals can be obtained in a region near the CF where data density from each radar is higher (square region shown in Fig. 2). Figures 6f, 7f, and 8f show UF, MF, and $\overline{w}$, respectively, for the square region. The UF, $\overline{w}$, and hence MF are improved especially for 3LR simulations, where

distances of nearest data are mostly less than 0.2 km (Figs. 2g and 2h). Although the profiles from 3XRSnap and 4SRSnap are improved as they capture the peak at middle altitude, the improvements are weaker than 3LR simulations at higher altitudes, where the distances of the nearest radar data points in the square region are similar as those from the entire domain for XR (Figs. 2g and 2h). It is suggested that the high data density, should be considered as an indicator of improved retrievals, as long as the scanning

the VCP is completed in 2 minutes.

      Increasing the number of Doppler radars in retrievals would reduce the uncertainties as analyzed by Bousquet et al. (2008) and North et al. (2017). Here we compare the 4SRSnap simulation with the 3LRSnap and 3XRSnap simulations. The 4SRSnap retrieval cannot significantly improve the UF$_5$ and UF$_{10}$ profiles compared to those from the 3XRSnap, as well as the number of updraft cores and $\overline{w}$ profiles,

and hence MF. Lower spatial resolutions of the C-SAPR VCP than the X-SAPR might induce more artifacts in the weaker updraft retrievals. The lower frequency radar (C-SAPR) can provide radar reflectivity measurements that may be easier to correct for hydrometeor and radome attenuation (e.g., Kurri and Huuskonen, 2008). In this case, it is perhaps advantageous to use the lower frequency radar to cover the domain sampled by the XSAPR network. However, if additional radars of the same or better

spatial resolution and VCP are available, the network architecture should be considered in order to maximize the triple-Doppler radar area by creating another sampling area with triple-Doppler radar observations.



### 3.3 Effect of VCP time period

The 2-min and 5-min time period VCP retrievals are compared to the snapshot retrievals to see how the VCP time periods affect the updraft retrievals. For the 3XR retrieval simulations, profiles of the number of updraft cores do not show significant differences among 3XRSnap, 3XR2min, and 3XR5min (Fig. 5b), consistent with little difference among those from WRFSnap, WRF2min, and WRF5min. This feature is also found in the 3LR simulations. However, some differences can be found in Figs. 6-8 showing updraft fractions, convective mass flux, and mean updraft. For both updraft threshold of 10 and 5 m s$^{-1}$, 3XR2min and 3XRSnap UF, $\overline{w}$, and hence MF are in close agreement at all altitudes and even with WRF output (WRFSnap and WRF2min) below 4.5 km, as well as with 3LR and 4SR simulations. The small impacts of 2-min time period are also found for the center square region (Figs. 6f, 7f, and 8f). For 3XR5min and 3LR5min simulations, however, UF$_{10}$, and $\overline{w}_{10}$ are significantly underestimated at 4-9 km AGL when compared to the snapshot retrievals (3XRSnap and 3LRSnap, respectively). The differences from the 3XR5min simulation result in significant underestimation of MF$_{10}$ at middle altitudes. These differences in UF and MF are also found even when comparing with the WRF UF/MF profiles averaged over 5 minutes (WRF5min). These features are common in 3XR, 3LR, and 4SR simulations. The comparison of UF$_5$, $\overline{w}_5$, and MF$_5$ for different time period from a given VCP show different features compared to those for the larger updraft threshold. As discussed in Sect. 3.2, the UF$_5$ profiles from the simulations are largely overestimated above 6 km and cannot resolve a peak at middle altitudes. The difference becomes larger for the 5-min VCP retrieval simulations. It is suggested that a longer VCP time period tends to underestimate areas of larger updrafts (> 10 m s$^{-1}$) and overestimate areas of weaker updraft (< 10 m s$^{-1}$). On the other hand, $\overline{w}_5$ from 3XR5min is underestimated above 5 km. These errors in UF$_5$ and $\overline{w}_5$ from 3XR5min produce large underestimation of MF$_5$ at middle altitudes and overestimation above 7 km. These features are also shown in 3LR5min and 4SR5min, but the underestimations of MF$_5$ at middle altitudes are small, since underestimation of $\overline{w}_5$ is relatively small for 3LR5min or overestimation of UF$_5$ is larger for 4SR5min.

Overall, the impacts from the 2-min VCP on the updraft retrieval can be small, whereas the 5-min VCP can significantly intensify uncertainties especially for stronger updraft regions above 6 km AGL. This is likely due to small convective evolution in 2 minutes while large evolution and advection in 5 minutes as





shown in Fig. 3. Potvin et al. (2012b) also showed a similar result that the data sampling in 3 minutes produced significant errors compared to shorter time period (1.5 min) and snapshot for supercell storms. Compared to the 3XR and 4SR retrievals for each VCP time period (2min and 5min), the 3LR2min and 3LR5min show better agreements.

**3.4     Effect of Advection Correction**

As presented in the previous section, the longer time VCPs more emphasize the uncertainties at upper levels. Because profiles of the updraft properties from WRF output do not change among the snapshot, 2-min average, and 5-min average, the differences found when comparing the simulated retrievals for 2-min and 5-min VCPs without advection correction and those for the snapshot VCPs are probably
associated with i) imposed advection and ii) cloud evolution, rather than time change of the updraft properties. Advection will move clouds and cause mismatch of cloud locations between PPI scans from different radars and even from the same radar. Meantime, cloud evolution cannot maintain the instantaneous cloud structures, resulting in observations of different cloud life stages by different PPI scans. Both issues result in deformation of cloud structures and may cause uncertainties in the wind
retrieval algorithm, especially the mass continuity assumption is not satisfied adequately. The cloud locations can be corrected using an algorithm proposed by Shapiro et al. (2010a) as described in Sect. 2.4.4. Here, we compare 2-min and 5-min VCP experiments to which the advection correction has been applied (2minadv, 5minadv) with those without the advection correction and snapshot experiments to see how the advection correction can improve the retrievals using 2-min and 5-min VCPs.

Figures 6e, 7e and 8e show UF, MF, and $\overline{w}$ profiles, respectively, from the 2-min and 5-min VCP 3XR simulations corrected for advection (3XR2minadv and 3XR5minadv, respectively), together with those from WRF snapshot and 3XRSnap. The advection-corrected retrievals for the 2-min VCP well improve these profiles as they are closer to the WRF2min profiles and even to the snapshot retrieval, while improvements are not significant for the 5-min VCP. Very similar improvements for the 2-min and 5-min
VCPs by advection corrections are found in 3LR simulations with advection correction (not shown).

Figure 9 shows comparisons of vertical cross sections between wind retrievals obtained before and after applying the advection correction for the updraft core shown in Fig. 3 (right column). Chosen vertical



cross sections go through the maximum updraft area at 7 km AGL. For the 2-min VCP retrievals, regions of updraft values > 5 m s$^{-1}$ are significantly corrected by the advection correction technique and maintain the top-left to bottom-right tilt of the WRF updraft structure. It is clear (Fig. 3 right column) that within 5 min the updraft structure has evolved not only in its tilt but also by the presence of a downdraft near its lower levels. Thus, when using a 5-min VCP a completely different updraft structure is reconstructed with different tilt and location of the maximum updraft velocity. The difficulty in improving the updraft retrieval using the advection correction, particularly for 5-min VCP, is likely due to fast evolution of convective clouds. The rapid evolution of the updraft structures simulated by the WRF are consistent with those from other modelling studies (e.g., Morrison et al., 2015; Hernández-Deckers and Sherwood, 2016) where the temporal evolution of the convective thermals can be significant over time periods larger than 2 min.

## 4 Summary and conclusions

Convective motions affect microphysical processes and control the transport of moisture, momentum, heat, trace gases and aerosols from the boundary layer to the upper troposphere. Accurate characterization of the convective transport requires vertical air velocity retrievals especially in the middle and upper part of convective cloud systems, and multi-Doppler radar networks have been used to probe convection and provide wind retrievals including vertical air motion estimates. While there is a plethora of studies illustrating the ability of multi-Doppler radar observations to capture the low-level wind divergence and circulation, there is little to show regarding the capability of this observing system to capture the upper level convective dynamics. This study addressed potential observational sources of errors in X-band triple-Doppler radar updraft retrieval using a sophisticated forward radar simulator (CR-SIM) with the WRF simulation output for an MCS on 20 May 2011 during the Midlatitude Continental Convective Clouds Experiment (MC3E) for a domain of 50 km × 50 km × 10 km and a three-dimensional variational (3DVAR) multi-Doppler radar-based wind retrieval technique (North et al., 2017). An extensive sensitivity analysis is conducted to investigate impacts of radar volume coverage pattern (VCP), the number of radars used for the multi-Doppler radar analysis, time periods for VCP (2 and 5 minutes), and advection correction. An advection correction technique proposed by Shapiro et al. (2010a) were applied




to the 2-min and 5-min VCP radar data. Updraft properties such as updraft fraction, mass flux, and updraft magnitude profiles with two different thresholds (5 m s$^{-1}$ and 10 m s$^{-1}$), from simulated multi-Doppler radar wind retrievals using three X-band Scanning ARM Precipitation Radars (X-SAPRs) are examined. The number of updraft cores are also investigated with a threshold of 5 m s$^{-1}$ at each height. The analysis results presented the following findings:

- As the previous literature pointed out, the updraft fraction profiles from the simulated wind retrievals suggested that the selected VCP elevation strategy and radar sampling volume resolution affect uncertainties in upper-level (~4.5 km) updraft retrievals using 3 X-SAPRs, and those uncertainties increase with height above 6 km AGL. In overall experiments in this study except the retrieval using the full grid radar data, stronger updrafts > 10 m s$^{-1}$ tend to be underestimated above 4.5 km, while areas of updrafts 5-10 m s$^{-1}$ are overestimated above 6.5 km. Those impact the retrieval of convective mass flux. These uncertainties are caused by low density and low resolution of radar data attributed to gaps between Plan Position Indicator (PPI) elevation angles and the radar sample volume increasing with distance from the radar.

- Increasing the maximum elevation angle and the density of the elevation angles of the radar VCP (i.e. 60° over elevation with 1° increment) can effectively improve the updraft retrieval, whereas an addition of data from a Doppler radar cannot significantly improve the updraft retrievals if the added radar VCP has inferior spatial resolutions.

- Shorter duration (2-min or less) radar VCPs are critical to producing high-quality vertical air motion retrievals using multi-Doppler based techniques. The 2-min VCP has small impacts on the snapshot updraft retrievals, but the 5-min VCP induces an important overestimation of areas of updrafts 5–10 m s$^{-1}$ above 6.5 km and underestimation of stronger updrafts > 10 m s$^{-1}$ at 4.5 – 8 km when comparing to the those obtained from the 5-min averaged WRF fields and even from the snapshot retrievals. Moreover, the areas of stronger updrafts (> 10 m s$^{-1}$) are overestimated above 8.5 km for the 5-min VCP.

- The advection correction works to improve the updraft fraction and mean updraft profiles as the profiles become closer to those from the snapshot retrievals and time averaged updraft fields, but it is still hard to improve stronger updraft retrievals especially for 5-min VCP. The magnitude of





improvement by the increase of elevation angles is larger than that by advection correction, even though the VCP needs 2 minutes. However, for the increasing elevations, which takes 5 minutes, the improvement is less than that from the original VCP completed within 2 minutes.

Gridding technique is also an important factor to determine the uncertainties in the wind retrievals. Sophisticated gridding techniques to cover the three-dimensional analysis domain at high spatial resolution, even for higher altitudes, tend to suppress the uncertainty (e.g., Majcen et al., 2008; Collis et al., 2010; North et al., 2017). Another error source that we did not consider in this study is hydrometeor fall speed estimate, which is generally estimated from radar reflectivity. The sophisticated attenuation correction techniques especially for shorter wavelength radars (e.g., Kim et al, 2008; Gu et al., 2011) and best estimates of hydrometeor fall speeds (Giangrange et al., 2013) are required to reduce the wind retrieval uncertainties.

In brief, the retrieval of the high-quality vertical velocities in the upper part of convective clouds is very challenging, while the multi-Doppler radar vertical velocity retrievals have been conventionally used to evaluate the CRM simulated dynamical fields. Some of the CRM simulations significantly overestimated compared to multi-Doppler radar vertical velocity retrievals (e.g., Varble et al., 2014; Fan et al., 2017). The present study would suggest that the multi-Doppler radar retrievals for MCSs tend to underestimate the updraft values at middle and upper levels and need to be carefully used considering the limitations of the radar observing system. Most of the improvements required in the sampling strategy of the observing system (higher maximum elevation angle, higher density elevation angles and rapid VCP time period) can be accomplished using rapid scan radar systems such as the DOW's or even phased array radar systems. However, even when such rapid scan radar networks are available, the multi-Doppler retrieval spatial domain will be fairly small compared to the entire radar network coverage. Despite of the limited domain, the observations do cover enough area to track isolated convective updrafts and contain enough samples to derive reliable, low-uncertainty estimates of updraft and downdrafts properties in convective clouds. Spaceborne radar systems with Doppler velocity capability such as the Earth Clouds Aerosols and Radiation Explorer (EarthCARE, Illingworth et al., 2015; Kollias et al., 2018b) or future spaceborne radar concepts (Tanelli et al., 2018) are expected to provide additional middle and upper level convective velocity observations especially over the tropical oceans.



# Acknowledgements

Portions of this work are funded by the U.S. DOE Office of Science's Biological and Environmental Research Program through the Atmospheric Radiation Measurement (ARM) and Atmospheric System Research (ASR) programs. P. Kollias is also supported by U.S. DOE grant DE‑ SC0012704. The
contribution of A. Shapiro has been supported by NSF grant AGS-1623626. The source code and user manual for the Cloud Resolving Model Radar Simulator (CR-SIM) are available at https://www.bnl.gov/CMAS/cr-sim.php.

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



**Tables and figures**

**Table 1: Simulated radar configurations and measurement strategy.**

|  | X-SAPR | C-SAPR |
|---|---|---|
| Radar frequency (GHz) | 9.5 | 5.5 |
| Beamwidth (degrees) | 1.1 | 1.0 |
| Number of elevation angles | 21 | 17 |
| Elevation angles (degrees) | 0.5, 1.5, 2.5, 3.5, 4.5, 5.5, 6.5, 7.5, 8.5, 9.5, 10.5, 11.5, 12.5, 14.0, 17.0, 20.0, 25.0, 30.0, 35.0, 40.0, 45.0 | 0.8, 1.2, 1.9, 2.6, 3.5, 4.4, 5.3, 6.4, 7.8, 9.6, 11.7, 14.3, 17.5, 21.4, 26.1, 33.0, 42.0 |
| Azimuth spacing (degrees) | 1.1 | 1.0 |
| Maximum observation range (km) | 40 | 120 |
| Range gate spacing (m) | 60 | 120 |
| Radar location | X-SAPRs (I4, I5, and I6) of Fig. 1 | C-SAPR I7 of Fig.1 |
| Antenna rotation rate* (° s$^{-1}$) | 28 | 18 |

5 * Antenna rotation rates used during the MC3E are presented and not used in this study.



**Table 2: Overview and short description of the different sensitivity simulations.**

| Simulation | Name | Specification |
|---|---|---|
| Control | 3FullGrid | i) All elevation angles from 3 X-SAPRs at each gridbox of the original WRF snapshot grid at 12:18:00 UTC  (no interpolation according to the radar beamwidth is considered |
| Radar VCP | 3XR | i) 3 X-SAPRs with 21 elevation angles ranging from 0.5 to 45 degrees over elevation angle |
|  | 3LRs | ii) 3 X-SAPRs with 60 elevation angles of 60 ranging from 0.5 to 59.5 degrees with equal increment of 1 degree |
|  | 4SR | iii) 4 radars including 3 X-SAPRs and the C-SAPR |
| Time period | Snap | i) Snapshot at 12:18:00 |
|  | 2min | ii) 2 minutes (6 snapshots) |
|  | 5min | iii) 5 minutes (15 snapshots) |
| Advection correction | (No name) | i) No advection correction |
|  | adv | ii) Advection correction proposed by Shapiro et al. (2010a) for time settings ii) and iii) |





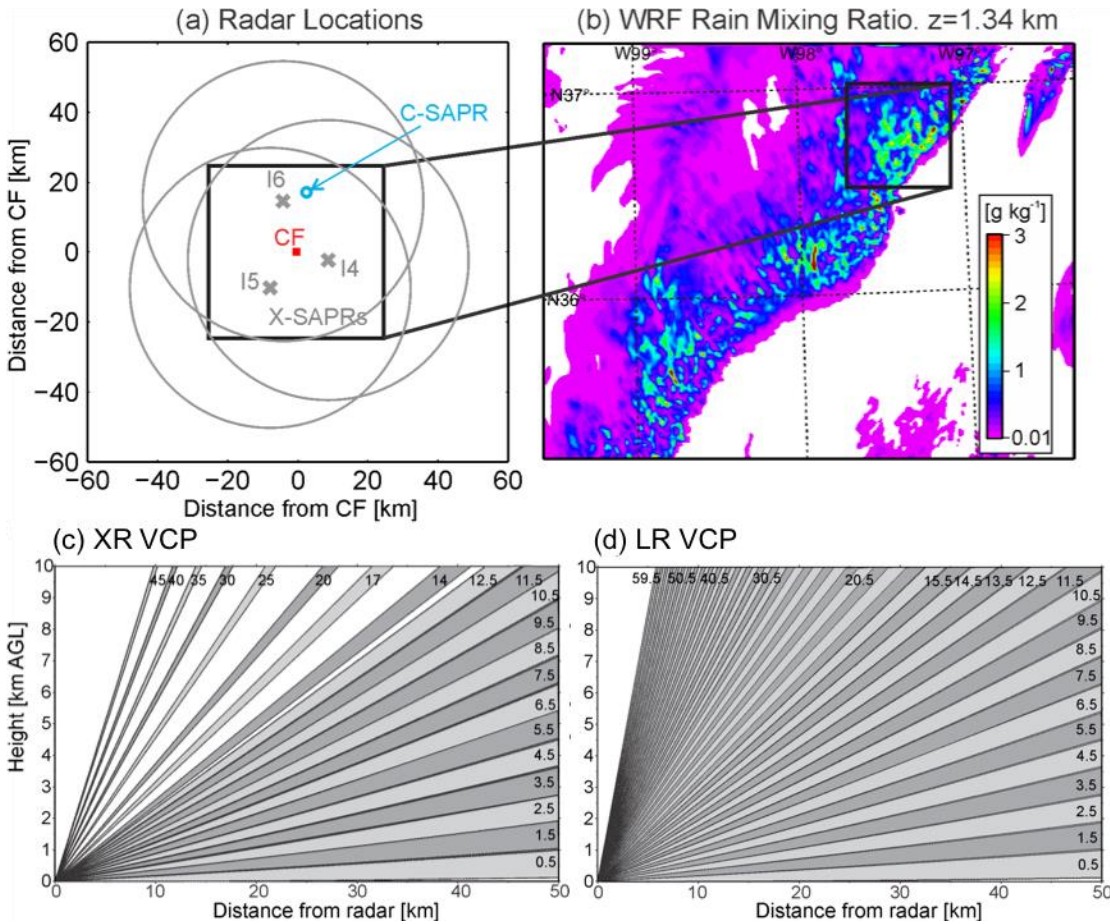

**Figure 1: (a) Locations of radars and the Department of Energy Atmospheric Radiation Measurements (ARM) Central Facility. Large gray circles represent maximum range of each X-band Scanning ARM Precipitation Radar (X-SAPR). (b) Rain water mixing ratios at 1.3 km altitude from the WRF simulation of a mesoscale convective system at 12:18:00 UTC on 20 May 2011. Black boxes represent the domain used for wind retrievals. (c and d) Elevation coverage for X-SAPR general VCP (XR) and high-density elevation volume coverage pattern (VCP) (LR), respectively.**



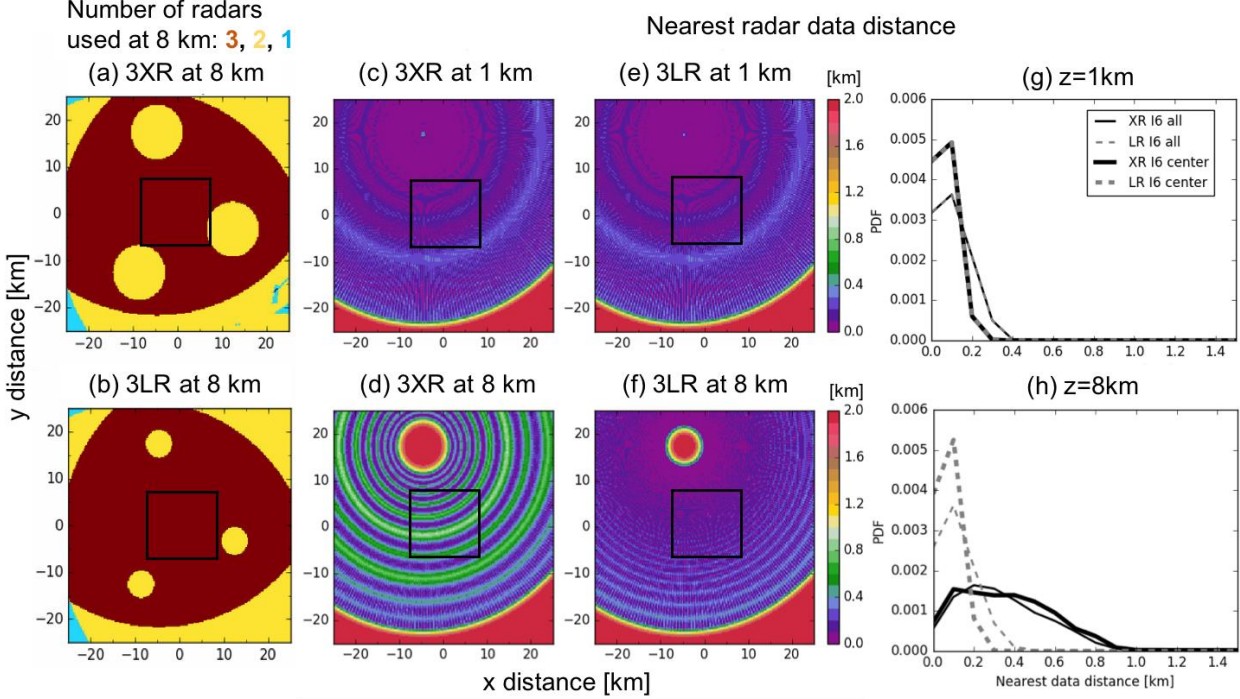

**Figure 2: (a and b) Number of radars used for retrievals at each grid box at 8 km above ground level (AGL) for 3XR VCP (a) and 3LR VCP (b), (c-f) distance of nearest radar data point at each grid box at 1 km (c and e) and 8 km (d and f) for the radar location of I6 with XR VCP (c and d) and LR VCP (e and f), and (g and h) histograms of the distance of nearest radar data point at 1 km (g) and 8 km (h) AGL normalized by the total number of data samples and the nearest distance bin size (0.1 km). In g and h, black sold lines represent the radar location of I6 with XR VCP, gray dashed lines represent the radar location of I6 with LR VCP, thin lines represent the entire horizontal domain, and thick lines represent a box area shown in (a-f).**



**Figure 3: (Left column) Horizontal distributions of X-band radar reflectivity at 7 km AGL from CR-SIM, (middle column) horizontal distributions of the WRF simulated vertical velocity at 7 km AGL, and (right column) vertical distributions of WRF-simulated vertical velocity along a line in the horizontal plots. Each row from top to bottom represents simulation time of 12:18:00, 12:19:00, and 12:22:00 UTC, respectively**.





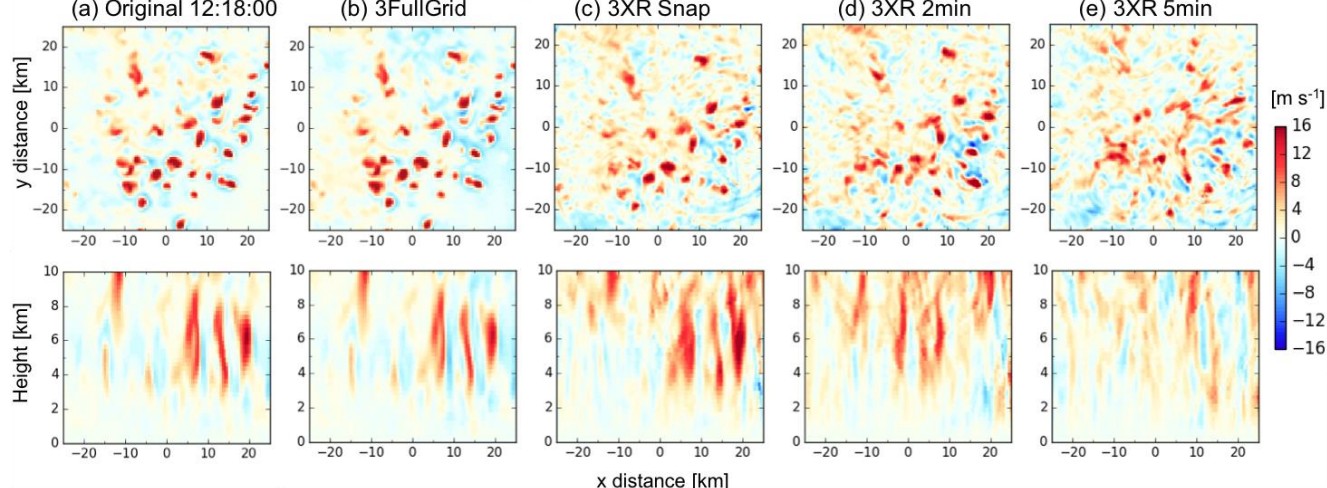

**Figure 4: (Top row) Horizontal distributions at 7 km AGL and (bottom row) vertical cross sections at y = 0 km of vertical velocity. Each column represents (a) the original WRF vertical velocity field and retrieved vertical velocity from (b) the 3FullGrid, (c) 3XRSnap, (d) 3XR2min, and (e) 3XR5min retrieval simulations.**




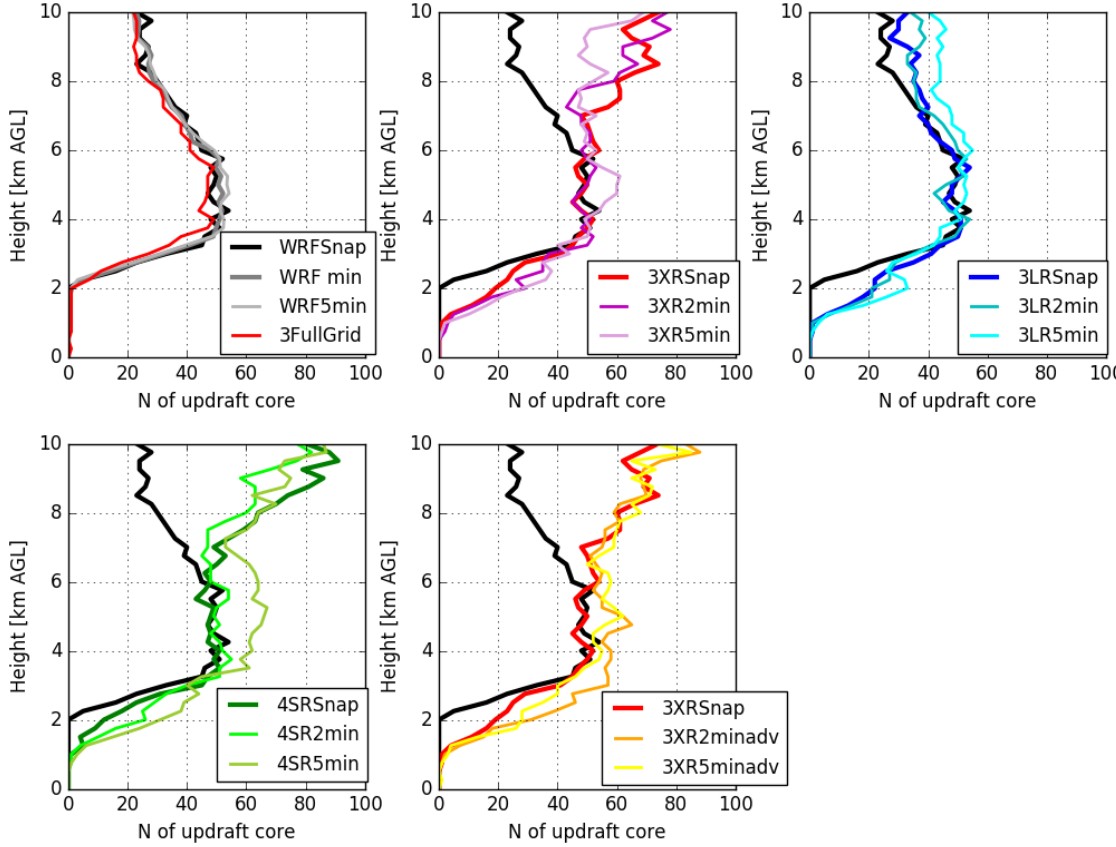

**Figure 5: Vertical profiles of the number of coherent updrafts with vertical velocity > 5 m s⁻¹. Colour represents different retrieval simulations as displayed in each panel. Dark gray line in (a) represents time average of the WRF output over 2 minutes from 12:18:00 to 12:19:40 UTC, and light gray in (a) represents time average of the WRF output over 5 min from 12:18:00 to 12:22:40 UTC. Each panel displays a profile from the WRF snapshot at 12:18:00 UTC by a black solid line.**





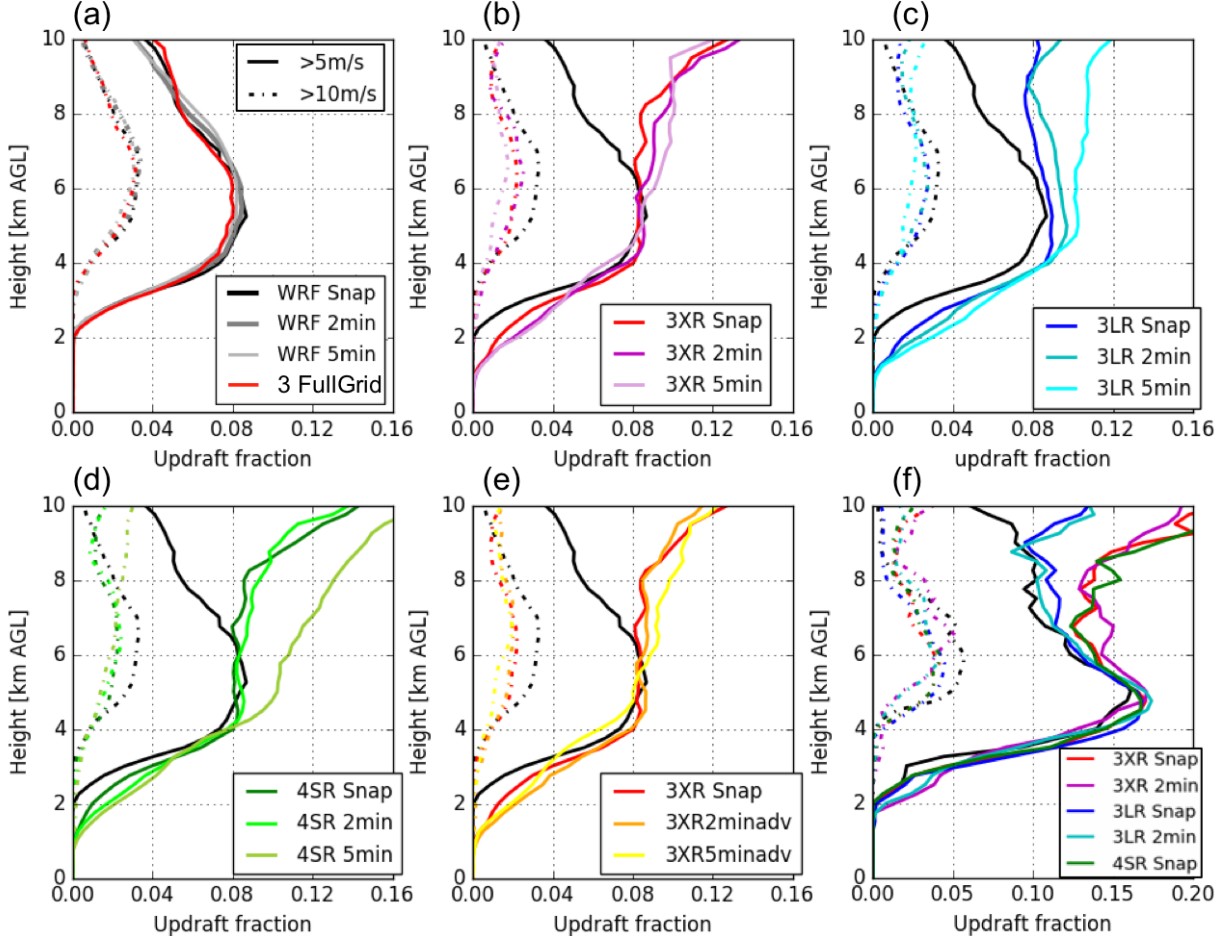

**Figure 6: Vertical profiles of updraft fractions with different thresholds of 5 m s⁻¹ (solid lines) and 10 m s⁻¹ (dashed lines) for the entire retrieval domain (a – e) and a center region shown as a box in Fig. 2 (f). Color represents different retrieval simulations as displayed in each panel.**







**Figure 7: Vertical profiles of convective mass flux with different updraft thresholds of 5 m s⁻¹ (solid lines) and 10 m s⁻¹ (dashed lines) for the entire retrieval domain (a–e) and a center region shown as a box in Fig. 2 (f). Color represents different retrieval simulations as displayed in each panel.**





**Figure 8: Same as Figure 7, but for mean updraft. Color representation is same as Figure 7.**



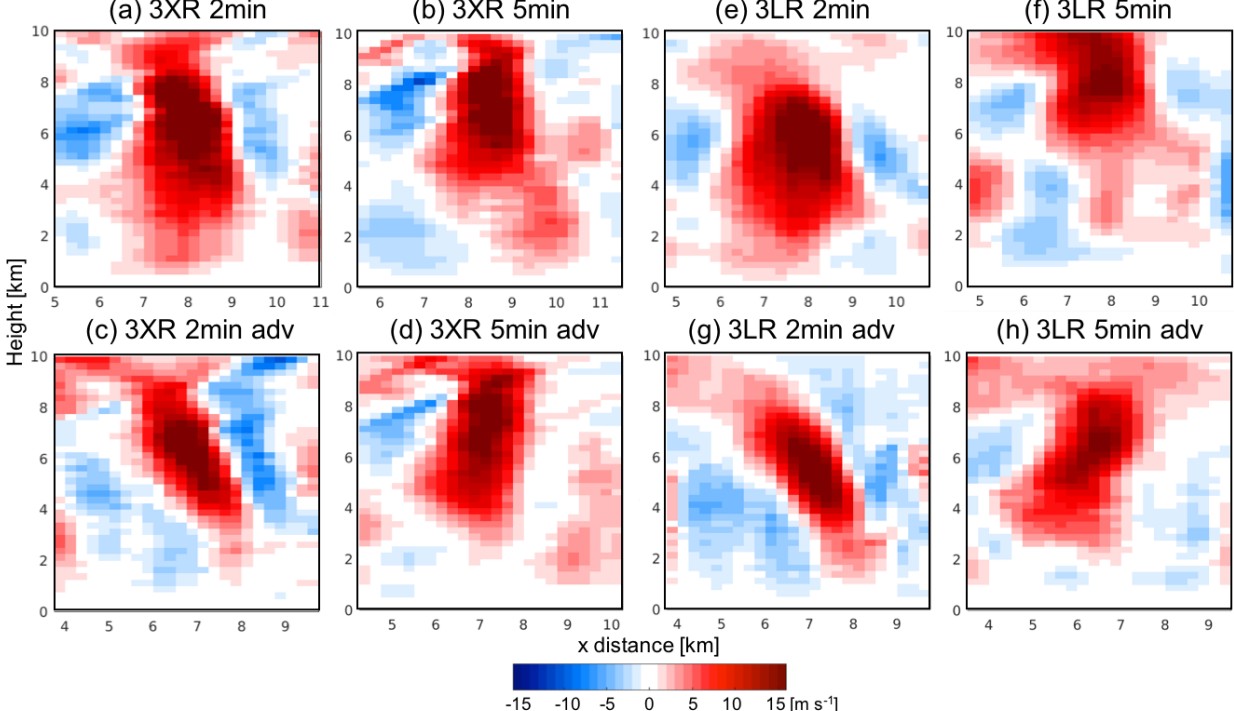

Figure 9: Vertical cross sections of vertical velocity for an updraft core from 3XR (a-d) and 3LR (e-h) wind retrieval simulations with 2-min and 5-min VCPs. Top raw and bottom raw display non-advection correction and advection-corrected retrievals, respectively. A selected updraft core is the same as shown in Fig. 3.