# Peer review of "Investigation of observational error sources in multi Doppler radar three-dimensional variational vertical air motion retrievals"

_Atmospheric Measurement Techniques, 2018_

## Referee Comment (RC1) · Anonymous Referee #1 · 29 Jan 2019

The manuscript is very well written and understandable. It shows a sensitivity analysis of the multi radar Doppler variational vertical wind velocity retrieval technique based on a simulated convective event as a function of the number of radar involved and their position, radar scan strategy and time sampling. Although most of the technical aspects are described by words or using citations, perhaps the Authors could evaluate to describe some parts in more formal details e.g. by adding appendix for example to describe the coupling of WRF outputs and electromagnetic simulations of backscattering cross section used in the manuscript. I recommend for publication after some minor revisions.

[Figure]

Main comments 1. Reflectivity weighted mean velocity: I am wondering if its calculation depends by the assumption made on the parameterization of the particle size distribution within the numerical weather model used. For example, if you assume two different WRF run one using a microphysical schemes 1 and a second independent run using a microphysical schemes 2 which is different by the previous scheme and assume that both microphysical schemes are constrained by the same mixing ratios for a given WRF grid point. Would you obtain two different reflectivity weighted mean velocity for the two assumed microphysical schemes? Am I right? Although I understand that within an OSSE scheme is not necessary reproduce the true (unknown) Doppler velocity from WRF outputs for a single radar, I would suggest the Authors to add some comments in this aspect. Is it worth performing a sensitivity test with respect to the particle size distribution assumption to understand if your simulated velocity fields are consistent with what we expect during actual observations?

2 In the advection correction section when you state: "The high temporal resolution WRF output allows us to evaluate the impact of advection and evolution of the cloud field during the time period needed to complete the radar network VCP." I am wondering if the 0.5 km horizontal resolution-WRF you are using resolves the processes involved within a time gap of 20 s or if 20s is just the time sampling used to write out the simulations. Later on when you state on pag 15: "…the number of coherent updrafts structures show little sensitivity to the VCP time. This can be attributed to the fact that the number of updraft coherent structures does not change within the 5 min required to complete all sampling strategies". Can it be attributed to the fact that you are not resolving processes at very short time scales although you have an output at such scales?

3. It would be probably nice to add a table in the paper that summarize the results quantitatively (e.g. RMS).

Minor: - page 5, lines 1-10: items 1-4: I am wondering if the gridding procedure (spherical to Cartesian) is introducing some errors and if the Authors took them into account.

[Figure]

Which is the interpolation method used? Is interpolation in step 3 really needed or it is something done for facilitating gradients calculations?

- page 5 line 13 and hereafter: maybe is "equivalent radar reflectivity factor".

- Eq. 1. It is not clear if you are applying the weights only in the horizontal plane or not. In other words, I was expecting that polar to Cartesian conversion was applied in 3D and not in 2D as Eq. 1 is suggesting. Please clarify.

Pag 12, line 17, "The corresponding plots for the latest model output (12:19:40 UTC) used to forward simulate the highest elevations of the 2-min VCP are shown in Fig. 3 (middle row)." Middle row of figure 3 shows 12:19:00 and not 12:19:40.

Pag 12 Advection correction section. What happen when you intercept the bright band in the advection correction scheme?

Fig4. May be I would add a third and fourth row of panels showing the differences between the various scenarios and the original one.

Figure 5. labels a, b,c, d, e are missing. Upper left panel: "2 min" is missing for the dark grey line

---

## Referee Comment (RC2) · Anonymous Referee #2 · 29 Jan 2019

**General comment**

The paper presents a sensitivity study aiming to address the error sources that affect the X-band Doppler radar-based retrieval techniques of vertical air motion. The main added-value of this work just lies in the comprehensive discussion of the limitations of such techniques. The paper productively contributes to add and extend the current research literature on this topic and can be accepted after some minor revisions.
Specific comments, suggestions and edits are provided below for the author's consideration and referenced by line number.

**Specific questions/issues**

- In my opinion, this study has one main limitation that needs to be considered and discussed. Such issue deals with the estimation of hydrometeors fall velocity ($V_f$). The authors state (Page 9, Lines 16-17) that $V_f$ used in their work is the one predicted by WRF model simulation; therefore, they assume that no errors related to $V_f$ are introduced in their experiment. In a more realistic scenario, the retrieval of wind field from radar Doppler measurements is strongly related to the variability of the terminal fall velocity of hydrometeors, which constitute a great source of uncertainty.
  I suggest to carry out, if possible, an additional experiment considering a scenario in which the hydrometeor fall speed is estimated from radar reflectivity measurements and not prescribed by WRF model or, at least, a more comprehensive discussion about the relationship between radar-estimated $V_f$ and wind retrieval.
- Pag. 9, lines 13-18: the attenuation along the path is one of the main issue affecting the quality of X-band radar measurements. In my opinion, the authors should carry out a more in-depth sensitivity analysis concerning this issue and its possible impact on vertical wind retrieval.
- The abstract should be more concise. I suggest to summarize the results in a four or five lines, at most.
- Section 2, paragraph 2.1: please add some more details, for the convenience of the reader, about the MCS event considered in this work and about the study area.
- Figure 2. Not very clear, in my opinion. Please avoid the use of jet colorbar in panel (c-f).
- The conclusion section should be reduced, by summarizing the main results of the study.
- The results of this study are presented only from a qualitatively perspective. Please introduce some common scores, such as the Root Mean Square Error, that quantitatively summarize the experiments performance.

**Technical corrections**

**Introduction**
- **Pag. 2, lines 13 and 20:** I suggest the use of the semicolon to improve the sentence structure.
- **Pag. 2, line 23:** please add "the" before "aforementioned".
- **Pag. 3, line 10 and Pag. 3, line 25:** please add a comma before "especially".
- **Pag. 3, lines 11-12:** I suggest to revise this sentence.
- **Pag. 3, line 13:** please add a comma before "by".
- **Pag. 3, line 16:** please replace "are" with "have been".
- **Pag. 4, line 8:** please add a comma before "that".

- **Pag. 4, line 11:** please replace "second" with "secondly".
- **Pag. 4, line 16:** please add a comma after "to do so".
- **Pag. 4. line 19:** please replace "we are investigating" with "we investigate".

**Data and methods**

- **Pag. 4, line 26:** I suggest to use "consists in" instead of "is composed" and to remove "following steps".
- **Pag. 5, line 20:** please substitute "retrieved" with "obtained" or "determined".
- **Pag. 6, lines 3-4**: remove "in their study".
- **Pag. 6, lines 5-6**: reformulate this sentence.
- **Pag. 7, line 3**: add a comma before "such".
- **Pag. 8, line 5**: add a comma before "with".
- **Pag. 8, line 8**: add a comma after "box".
- **Pag. 9, line 10**: please add "carried out" before "in this study".
- **Pag**. **13, lines 10-11**: reformulate this sentence.

**Results**

- **Pag. 13, line 14 and line 15**: add a comma after "fields" and before "field", respectively.
- **Pag. 14, line 13**: add a comma before "using the original...".
- **Pag**. **16, line 18**: add a comma before "but".
- **Pag. 16, line 26**: please revise "velocities lather than".
- **Pag. 18, line 14**: please remove the comma between "density" and "should".
- **Pag. 21, line 5**: please add a comma after "VCP".
- **Pag. 21, line 9**: add a comma after "…2016)".

**Conclusions**

- **Pag. 21, line 27**: please replace "were" with "was".

---

## Referee Comment (RC3) · Anonymous Referee #3 · 30 Jan 2019

**Overarching Comments**

This paper describes a rigorous evaluation of the North et al. (2017) 3DVAR multi-Doppler air motion retrieval using a high resolution cloud resolving model and a radar-model simulator and carefully examining selected common sources of uncertainty: characteristics of the scanning strategy, time updates of the scanning strategy, and advection corrections to the data. The novel aspect of the paper is that the experiments can control for many aspects of the sources of uncertainty given the high temporal resolution of the model output which can be subsampled nearly arbitrarily for comparison with the retrievals. The paper makes the case that for

some characteristics of estimating vertical velocity in triple-Doppler networks, that having a Doppler radar capable of scanning a PPI in less than 2 minutes is desirable, particularly for accurately estimating the characteristics of weak updrafts.

One major comment I have is that while the paper's methodology is thorough and the figures are clear and illustrative, the presentation of the results is done in an illogical fashion and does not allow for an easy read. The text refers to many multi-panel figures in a repeated fashion that lead the reviewer to do a lot of page flipping. I suggest breaking up the Figures 5, 6, 7, and 8 into different figures that follow the logical flow of Section 3 without having to refer back to the figure subpanels over and over again. Alternatively, discussion of the variables in Figs. 5-8 could be done sequentially by retrieved variable rather than by observing strategy. I believe that the former strategy would be easier for the authors to do.

I believe the paper is acceptable for publication with these minor revisions in mind.

**Specific Comments**

1. Should the title be revised to include *3DVAR* retievals? The paper does not include other traditional dual-Doppler retrievals such as in CEDRIC (which have different assumptions about integrating the continuity equation than 3DVAR), or more advanced analyses such as in SAMURAI (Bell et al.). Different techniques could fare better or worse than the 3DVAR technique.

2. It should be stated in the methdology that this analysis is based on a single case, and performance assessment could vary for different storm characteristics and propagation speeds. For example, a slow moving tropical squall line in low

vertical wind shear could present less of a challenge for multi-Doppler retrievals than the highly-sheared May 20 MC3E squall line.

3. P5, L3, and elsewhere: the apostrophe is unnecessary.

4. P5, L25: suggest removing the word 'at'.

5. P6, L15-16: the CRSIM simulates the phase shift and the differential phase shift, not the phase and specific differential phase, correct?

6. P7, L8: An analysis domain top of 10 km does not top the storm, correct? This would not allow for correct estimation of the storm-top divergence properly in the interpolated simulated radar data. Does this lead to some of the uncertainty unnecessarily?

7. P8, L20: "not unfolded correctly observed Doppler velocity" needs rewording.

8. P9, L14: "at" should be *in*.

9. P13, L2: The tilt would depend on the propagation speed and the vertical wind shear, which in some regimes (especially in the tropics) could be smaller than in this case.

10. P13, L11: The smoothness function is a Cressman or Barnes filter? If not, what equation did you use?

11. P14, L1: Suggest inserting "potentially" before "large".

12. P14, L25-26: "...but it tends to have higher uncertainty in the areas around the location of strong convection..." - this seems highly subjective. Can you make a quantitative statement related to this topic?

13. P15, L7: Regarding an updraft core defined as being larger than 0.5 km$^2$: Is this at a single level? This is one single pixel, correct? Since the retrieval has a resolution that is effectively 4-6 times the grid spacing, would one expect it to retrieve such small updraft features? Perhaps it is not surprising then that the retrievals have such issues resolving updrafts (particularly weak ones) that are single pixels in the grid? Perhaps increasing this threshold to have larger-sized updrafts (say larger than 2-3 km$^3$) would reveal different uncertainties.

14. P15, L16: Suggest inserting an "A" before "Noticable".

15. P16, L8-10: This sentence is awkward.

16. P20, L12-13: "cloud evolution cannot maintain the instantaneous cloud structures": Can you be more specific what you mean by this?

17. P22, L6: "As previous literature has pointed out..."

18. P22, L8: 4.5 km is mid-levels, no?

19. P22, L16: Need comma after "i.e."

20. P22, L18: "inferior" should be "lower"?

21. P22, L28: Suggest replacing "hard" with "challenging".

22. P23, L10: Scott G.'s last name is misspelled.

23. P23, L20: "DOW's" should be "Doppler on Wheels mobile radars"

———————————

---

## Referee Comment (RC4) · Anonymous Referee #4 · 1 Feb 2019

This paper investigates the impact of some sources of uncertainty on multiple-Doppler analysis of ground based radar data, with emphasis on vertical velocity. It is well written, clear and easily understandable. In fact, I don't have much to say about the points raised by the authors. The analysis process is good and the scientific argumentation is excellent.

Having said that, however, I am a little sceptical about the overall impact of this work and its general interest for the radar user community. Without being rude, it seems quite obvious that improving the VCP, the number of radars, or the time sampling resolution will have a positive impact on retrieved wind fields and, in particular, vertical velocity.

[Figure]

This has been shown previously by many authors in many different radar network configurations. While I agree that findings and recommendations resulting from this study would be useful for ARM RGP laboratory users, they would be difficult to apply to other networks. Furthermore using numerical model outputs and radar simulators to build a reference wind field is quite common nowadays and cannot be considered as a new concept.

Actually reading section 2.3 of the paper should generate quite some frustration among any scientist interested in radar wind retrieval. Indeed, questions regarding the impact of prescribed hydrometeor fall speed, potential masks, attenuation by rainfall (especially at X-band), or velocity folding count among the main sources of questioning for users and developers of multiple-Doppler analysis methods. To the best of my knowledge, these issues have never really been addressed to date and I was hoping that this study would help to clarify them, which would have contributed to make this paper a truly original contribution to the field. To be consistent with my remark, however, I must mention that the study on the impact of advection is original and does answer important questions. .

To sum up, the work presented by the authors is of good quality, but its contribution to the field with respect to previous studies seems quite poor to me and results are barely applicable to networks other than ARM RGP laboratory. From there I see two options: 1/ Improving the paper by investigating additional sources of uncertainty such as Vt, velocity folding, attenuation... among others, or 2/ Clearly state that this study aims to optimize the performance of the ARM RGP network, without seeking to generalize its findings. Option 1 would require substantial additional work, but would undoubtedly represent an important contribution to the field. Option 2 would mostly imply cosmetic work (title, introduction, conclusions), but the impact of this paper would be limited.

Other remarks:

1/ Results are based on a single case. Authors should keep that in mind in their conclusions. The effectiveness of vertical wind retrieval depends on many factors, including wind shear for example.

2/ More details are needed regarding the wind retrieval method used in this paper. What about data interpolation and air-mass continuity equation (e.g., boundary conditions) ?

3/ More details are needed about the investigated weather system. Authors should include additional material to better describes the overall structure of the storm (e.g. vertical cross-section of model/radar reflectivity/wind fields).

---

## Referee Comment (RC5) · Anonymous Referee #3 · 14 Feb 2019

I believe the authors have done a good job revising the manuscript. Thanks for making your paper improved.

---

## Author Comment (AC1) · 14 Feb 2019

**Responses to comments from Referee #1**

Thank you very much for your comments and suggestions. Providing this valuable feedback has helped to improve the current manuscript. We have modified the manuscript, taking into account all referee's suggestions. The following contains our detailed responses to your comments, with our responses in plain type given underneath your original comments in bold type.

**The manuscript is very well written and understandable. It shows a sensitivity analysis of the multi radar Doppler variational vertical wind velocity retrieval technique based on a simulated convective event as a function of the number of radar involved and their position, radar scan strategy and time sampling. Although most of the technical aspects are described by words or using citations, perhaps the Authors could evaluate to describe some parts in more formal details e.g. by adding appendix for example to describe the coupling of WRF outputs and electromagnetic simulations of backscattering cross section used in the manuscript. I recommend for publication after some minor revisions.**

Thank you for the referee for the appreciation. The radar simulator software and its user's guide are publicly available at https://you.stonybrook.edu/radar/research/radar-simulators/. The user's guide includes detailed descriptions about how the WRF output is coupled in the simulator and what electromagnetic scattering assumption is used. We would like to skip repeating the descriptions in the manuscript. Moreover, we are preparing to publish a separate paper to introduce the simulator and its applications.

Here we briefly describe about how the WRF output is coupled and the scattering assumption for the referee's reference.

The CR-SIM forward simulator is tailored to compute radar observables by integrating scattering properties over particle size distributions (PSDs) for each hydrometeor, on the basis of microphysics schemes incorporated in the weather model simulation. The environmental variables are obtained from the WRF model output variables and/or calculated using the obtained variables if necessary. Scattering properties of single particles are calculated using the T-matrix method and packaged as look-up tables (LUTs) in the simulator. The simulated 'idealized' observation variables are provided at each gridbox of the input model grid.

The LUTs compose the computed complex scattering amplitudes for equally spaced particle sizes using a T-matrix method developed by Mishchenko and Travis (1998) and Mishchenko (2000). The LUTs for each hydrometeor class corresponding to the model simulation data (e.g., liquid cloud, rain, snow, cloud, ice, graupel) are pre-built by setting particle phase, particle bulk density, and particle aspect ratio. For each hydrometeor class, the complex scattering amplitudes are pre-calculated for the total of 91 elevation angles from 0° to 90° with a spacing of 1°, different temperature ranges for liquid hydrometeors, different possibilities of particles densities for solid hydrometeors, and few different assumptions about particle aspect ratios.

**Main comments**
**1. Reflectivity weighted mean velocity: I am wondering if its calculation depends by the assumption made on the parameterization of the particle size**

distribution within the numerical weather model used. **For example, if you assume two different WRF run one using a microphysical schemes 1 and a second independent run using a microphysical schemes 2 which is different by the previous scheme and assume that both microphysical schemes are constrained by the same mixing ratios for a given WRF grid point. Would you obtain two different reflectivity weighted mean velocity for the two assumed microphysical schemes? Am I right? Although I understand that within an OSSE scheme is not necessary reproduce the true (unknown) Doppler velocity from WRF outputs for a single radar, I would suggest the Authors to add some comments in this aspect. Is it worth performing a sensitivity test with respect to the particle size distribution assumption to understand if your simulated velocity fields are consistent with what we expect during actual observations?**

Parameterizations of the calculation of the reflectivity-weighted mean fall velocity (e.g., particle size distribution and $V_f$-D relationship) depend on the assumptions in the microphysical scheme used in the WRF simulation. The referee is right; if we use data from different WRF simulation coupled with a different microphysics scheme, the reflectivity-weighted mean fall velocity calculation follows the parameterizations of the microphysics scheme used. Therefore, the calculated reflectivity-weighted mean fall velocity could be different even if the simulated mixing ratio was same.

We agree with the referee that a sensitivity test to see an impact of particle size distribution assumption on the simulated Doppler velocity field is worthwhile. The present study, however, focused on uncertainties attributed to radar observation sampling in the retrieved wind field, rather than in an assessment of uncertainty of observed Doppler velocity due to microphysics by using the forward simulator. We would like to include the sensitivity test to a separate paper where we study impacts of hydrometeor particle assumptions, such as size distribution, terminal velocity, bulk density, and aspect ratio, on the simulated radar variables (reflectivity, Doppler velocity, and polarimetric variables). Thank you for the valuable suggestion.

We have tested a sensitivity of hydrometeor fall speed estimates to the vertical velocity retrieval. Please see a response to the referee #2's second comment.

**2 In the advection correction section when you state: "The high temporal resolution WRF output allows us to evaluate the impact of advection and evolution of the cloud field during the time period needed to complete the radar network VCP." I am wondering if the 0.5 km horizontal resolution-WRF you are using resolves the processes involved within a time gap of 20 s or if 20s is just the time sampling used to write out the simulations. Later on when you state on page 15: "…the number of coherent updrafts structures show little sensitivity to the VCP time. This can be attributed to the fact that the number of updraft coherent structures does not change within the 5 min required to complete all sampling strategies". Can it be attributed to the fact that you are not resolving processes at very short time scales although you have an output at such scales?**

The WRF simulation was performed with a time step of 2 seconds and output was saved every 20 seconds. We believe that the WRF simulation captured short time evolution well. Since we used outputs every 20 seconds in the analysis, the radar simulator analysis in this study might not capture the dissipation and formation of updraft cores within 20 seconds. Although it is possible that a few of the updraft cores dissipated or formed within 5 minutes, as we can track individual updraft cores using the

20-sec outputs (e.g. Fig. 3), the evolution of updraft cores was captured by the 20-sec outputs, and the number of updraft cores did not change significantly within 5 minutes.

**3. It would be probably nice to add a table in the paper that summarize the results quantitatively (e.g. RMS).**

We calculated root mean square errors (RMSEs) of UF, MF, and $\bar{w}$ profiles for the updraft thresholds 5 m s$^{-1}$ and 10 m s$^{-1}$ profiles above 2 km AGL for all experiments and added Table 3 to present those values in the revised manuscript.

**Minor:**
**- page 5, lines 1-10: items 1-4: I am wondering if the gridding procedure (spherical to Cartesian) is introducing some errors and if the Authors took them into account. Which is the interpolation method used? Is interpolation in step 3 really needed or it is something done for facilitating gradients calculations?**

Because we created radar polar coordinate datasets to emulate the radar sampling strategies, the gridding is mandatory before applying the 3DVAR wind retrieval used in this study. We interpolated the created radar polar coordinate data into the Cartesian grid using the Barnes distance-dependent weighting technique (Eq. 1 of the manuscript). But, for the 3FullGrid simulation, the gridding procedure was not applied because we used the original grid.

The referee is correct. The gridding process can also be an error source in the vertical velocity retrieval. This error can be included and seen when we compare the 3FullGrid simulation with the other simulations using radar VCPs. We discussed about this in the second paragraph of Section 3.1 in the revised manuscript. We used fixed settings for the gridding for all radar simulations except 3FullGrid in the manuscript, and therefore the differences between the radar simulation results (except 3FullGrid) may not include uncertainties attributed to the settings for gridding process.

**- page 5 line 13 and hereafter: maybe is "equivalent radar reflectivity factor".**

Done. We used a word "equivalent radar reflectivity factor" or "$Z$" in the revised manuscript.

**- Eq. 1. It is not clear if you are applying the weights only in the horizontal plane or not. In other words, I was expecting that polar to Cartesian conversion was applied in 3D and not in 2D as Eq. 1 is suggesting. Please clarify.**

We used this equation for both horizontal and vertical interpolations for gridding. We added the following sentence after the equation: "The equation was applied in both horizontal and vertical interpolations."

**Pag 12, line 17, "The corresponding plots for the latest model output (12:19:40 UTC) used to forward simulate the highest elevations of the 2-min VCP are shown in Fig. 3 (middle row)." Middle row of figure 3 shows 12:19:00 and not 12:19:40.**

We replaced the plots in the middle raw with those at 12:19:40 UTC and revised the last sentence of the Section 2.4.3 to read "The corresponding plots for the 13th model output (4 minutes after the first scan, 12:22:00 UTC) used to forward simulate the highest elevations of the 5-min VCP simulations is shown in Fig. 3 (bottom row)." Moreover, we changed a color scale of the reflectivity plots to clarify convective cores. Thank you for pointing this out.

**Pag 12 Advection correction section. What happen when you intercept the bright band in the advection correction scheme?**

The advection correction procedure has not been tested in the presence of a bright band. We can only speculate that there is nothing particular about a bright band that would cause special difficulties, because the advection correction technique does not use information that may be difficult to obtain information in bright bands (e.g., the detailed nature of the scatterers or the terminal velocities). If bright bands are horizontally inhomogeneous, the spatial variations should be able to be "tracked" with the advection correction technique as other reflectivity features are tracked. However, if there is a persistent largely homogeneous bright band covering the entire horizontal domain, the bright band might cause some problems.

In the present study, the radar simulator did not consider melting hydrometeors, thus a bright band was not distinguished in the simulated radar reflectivity fields. Therefore the advection correction was not affected by an issue that is caused by a presence of a bright band, if any.

**Fig4. May be I would add a third and fourth row of panels showing the differences between the various scenarios and the original one.**

We agree with the referee. However, because the horizontal grid box size is different between the WRF data (0.5 km) and the simulated retrieval data (0.25 km), we decided not to include the horizontal cross sections of the difference for the simulated retrieval from the WRF data. Instead, we added RMSEs of the profiles in Table 3 as suggested in the previous comment.

**Figure 5. labels a, b,c, d, e are missing. Upper left panel: "2 min" is missing for the dark grey line.**

Done.

---

## Author Comment (AC2) · 14 Feb 2019

**Responses to comments from Referee #2**

Thank you very much for your comments and suggestions. Providing this valuable feedback has helped to improve the current manuscript. We have modified the manuscript, taking into account all referee's suggestions. The following contains our detailed responses to your comments, with our responses in plain type given underneath your original comments in bold type.

**General comment**
**The paper presents a sensitivity study aiming to address the error sources that affect the X-band Doppler radar-based retrieval techniques of vertical air motion. The main added-value of this work just lies in the comprehensive discussion of the limitations of such techniques. The paper productively contributes to add and extend the current research literature on this topic and can be accepted after some minor revisions.**

Thank you for the referee's appreciation for the study.

**Specific questions/issues**
**In my opinion, this study has one main limitation that needs to be considered and discussed. Such issue deals with the estimation of hydrometeors fall velocity (Vf). The authors state (Page 9, Lines 16-17) that Vf used in their work is the one predicted by WRF model simulation; therefore, they assume that no errors related to Vf are introduced in their experiment. In a more realistic scenario, the retrieval of wind field from radar Doppler measurements is strongly related to the variability of the terminal fall velocity of hydrometeors, which constitute a great source of uncertainty. I suggest to carry out, if possible, an additional experiment considering a scenario in which the hydrometeor fall speed is estimated from radar reflectivity measurements and not prescribed by WRF model or, at least, a more comprehensive discussion about the relationship between radar-estimated Vf and wind retrieval.**

We agree with the referee that the hydrometeor fall velocity estimate can be a source of uncertainty. We have tested an impact of Vf estimate using reflectivity-based mass-fall velocity relationships proposed by Caya (2001) on the retrieved vertical velocity (presented below). The relationships generally tend to produce slower fall speeds than the reflectivity-weighted mean fall velocities calculated for the WRF simulation case. The retrieved vertical velocity areas from the simulation using the relationships were underestimated. For the referee's reference, we show a comparison between the simulated reflectivity-weighted fall velocity and a relationship for liquid from Caya (2001) and a comparison of the updraft fraction (UF) profiles with the WRF simulation and the 3FullGrid simulation using the relationship. However, the result does not necessarily mean that the relationships were incorrect or the retrieval was failure. It is hard to say whether the fall velocity estimate from these relationships or the hydrometeor fall velocity predicted by model simulation is more reliable. The present study focused on the uncertainties attributed to radar observation sampling, and we decided not to include the sensitivity test of the hydrometeor fall velocity estimate.

[Figure]

- • Simulated reflectivity-weighted fall velocity
- – reflectivity-based hydrometeor fall speed relationship for liquid by Caya (2001)

Fig. R1: (a) Simulated reflectivity-weighted Vf versus reflectivity based on the Morrison 2-moment microphysics scheme at a height of 1 km (blue dots) and a reflectivity-based mass-fall velocity relationships for liquid proposed by Caya (2001) (orange line). (b) Vertical profiles of updraft fractions with different thresholds of 5 m s$^{-1}$ (solid lines) and 10 m s$^{-1}$ (dashed lines). In (b), Black lines represent the WRF snapshot at 12:18:00 UTC, red lines represent the 3FullGrid simulation, and blue lines also represents the 3FullGrid simulation, but used the hydrometeor fall velocity estimates proposed by Caya (2001). The reflectivity-based mass-fall velocity relationships for liquid and ice hydrometeors were parameterized:

$$V_f = 5.94M^{0.125}\exp(\tfrac{h}{20}) \text{ for liquid}$$

$$V_f = 1.15M^{0.083}\exp(\tfrac{h}{20}) \text{ for ice}$$

$$M = \exp(\frac{Z - 43.1}{7.6})$$

where $h$ represents height in km, and $Z$ represents reflectivity in a logarithmic scale.

**Pag. 9, lines 13-18: the attenuation along the path is one of the main issue affecting the quality of X-band radar measurements. In my opinion, the authors should carry out a more in-depth sensitivity analysis concerning this issue and its possible impact on vertical wind retrieval.**

The attenuation for hydrometeors can significantly impact the radar reflectivity measurements, but little impact on the Doppler velocity measurements. However, the attenuation in the reflectivity field can induce underestimations in the reflectivity-based hydrometeor fall velocity estimates. The underestimated hydrometeor fall velocity estimates could induce underestimation of the vertical velocity as shown in the response to the previous comment.

**The abstract should be more concise. I suggest to summarize the results in a four or five lines, at most.**

We itemized the results in the abstract and reduced the abstract.

**Section 2, paragraph 2.1: please add some more details, for the convenience of the reader, about the MCS event considered in this work and about the study area.**

We added short descriptions about the observed MCS and WRF-simulated MCS to the first paragraph of Section 2 and Section 2.1, respectively. For the observed MCS, we added: "This squall-line MCS was oriented in northeast-southwest direction extending for approximately 1000 km (Fan et al., 2017). The convective region had approximately 50 km width and trailed a distinct stratiform precipitation area when it passed through the ARM SGP site from 09:20 UTC to 11:40 UTC."   For the WRF-simulated MCS, we added: "The simulated MCS comprised a convective precipitation region at the leading edge of the system and a stratiform precipitation trailed by the convective region, as similar as the observation. The MCS passed through the ARM SGP radar observation site approximately one hour later than the observation (at around 12:18 UTC), and a stronger convective precipitation region formed slightly (~20 km) to the north of the ARM SGP site."

**Figure 2. Not very clear, in my opinion. Please avoid the use of jet colorbar in panel (c-f).**

We changed the color scale for the nearest neighbor distance plots (c-f) in Fig. 2.

**The conclusion section should be reduced, by summarizing the main results of the study.**

We tried to reduce the amount of the itemized results by removing duplicated sentences.

**The results of this study are presented only from a qualitatively perspective. Please introduce some common scores, such as the Root Mean Square Error, that quantitatively summarize the experiments performance.**

We calculated root mean square errors (RMSEs) of UF, MF, and $\bar{w}$ profiles for the updraft thresholds 5 m s$^{-1}$ and 10 m s$^{-1}$ profiles above 2 km AGL for all experiments and added Table 3 to present those values in the revised manuscript.

**Technical corrections**
**Pag. 2, lines 13 and 20: I suggest the use of the semicolon to improve the sentence structure.**

Done. We rephrased the sentence at line 20 as "One drawback of profiling radar techniques is their limited sampling of individual storms and the lack of information on the temporal evolution of the convective dynamics and structure; the observational limitations, thus, make the use of the techniques in model evaluation challenging."

**Pag. 2, line 23: please add "the" before "aforementioned".**

Done.

**Pag. 3, line 10 and Pag. 3, line 25: please add a comma before "especially".**

Done.

**Pag. 3, lines 11-12: I suggest to revise this sentence.**

We revised this sentence to read "Clark et al. (1980) estimated errors attributed to cloud evolution in horizontal and vertical wind estimates from multiple Doppler radar measurements." Thank you for pointing this out.

**Pag. 3, line 13: please add a comma before "by".**

Done.

**Pag. 3, line 16: please replace "are" with "have been".**

Done.

**Pag. 4, line 8: please add a comma before "that".**

We separated this sentence into the following two: "It is possible that some of the errors are associated with radar volume coverage pattern strategy that does not satisfy the requirement for high spatiotemporal observations." and "This issue has been highlighted in recent studies with high-resolution CRM simulations of convective cloud properties (e.g., Morrison et al., 2015; Hernández-Deckers and Sherwood, 2016)."

**Pag. 4, line 11: please replace "second" with "secondly".**

Done.

**Pag. 4, line 16: please add a comma after "to do so".**

Done.

**Pag. 4. line 19: please replace "we are investigating" with "we investigate".**

Done.

**Pag. 4, line 26: I suggest to use "consists in" instead of "is composed" and to remove "following steps".**

Done.

**Pag. 5, line 20: please substitute "retrieved" with "obtained" or "determined".**

Thank you for the suggestion. We used "obtained" instead of "retrieved."

**Pag. 6, lines 3-4: remove "in their study".**

Done.

**Pag. 6, lines 5-6: reformulate this sentence.**

We rephrase this sentence as "This case has been analyzed for its dynamical and microphysical structures by many previous studies" and moved it to the first paragraph of Section 2.

**Pag. 7, line 3: add a comma before "such".**

Done.

**Pag. 8, line 5: add a comma before "with".**

Done.

**Pag. 8, line 8: add a comma after "box".**

Done.

**Pag. 9, line 10: please add "carried out" before "in this study".**

Done.

**Pag. 13, lines 10-11: reformulate this sentence.**

The advection correction procedure seeks to minimize a cost function that contains the frozen turbulence constraint and terms that confer spatial smoothness on the pattern-translation components. Appropriate values of the coefficient of the spatial smoothness terms depend on the horizontal grid spacing and a typical value of the tracked variable in the case. Based on preliminary tests (not shown), we deemed a coefficient of 300 dBZ^2 to be acceptable.

We added a following phrase to the previous paragraph:

"The advection correction procedure seeks to minimize a cost function that contains the frozen turbulence constraint and terms that confer spatial smoothness on the pattern-translation components."

and revised the sentence to read:

"A weighting coefficient of the spatial smoothness terms in the cost function coefficient depends on the analysis grid spacing and the structure of the field being advected. An appropriate value of the coefficient can be determined by running some sensitivity tests. Based on preliminary tests (not shown), we deemed a coefficient of 300 dBZ^2 to be acceptable."

**Pag. 13, line 14 and line 15: add a comma after "fields" and before "field", respectively.**

We added a comma after "fields" and "field", respectively. Thank you for the suggestion.

**Pag. 14, line 13: add a comma before "using the original...".**

Done.

**Pag. 16, line 18: add a comma before "but".**

Done.

**Pag. 16, line 26: please revise "velocities lather than".**

We revised it to read "updraft values."

**Pag. 18, line 14: please remove the comma between "density" and "should".**

Done.

**Pag. 21, line 5: please add a comma after "VCP".**

Done.

**Pag. 21, line 9: add a comma after "…2016)".**

To specify the previous studies, we avoided to add a comma there. Instead, we revised this sentence to read "The rapid evolution of the updraft structures simulated by the WRF are consistent with those from other modelling studies where the temporal evolution of the convective thermals can be significant over time periods larger than 2 min (e.g., Morrison et al., 2015; Hernández-Deckers and Sherwood, 2016)."

**Pag. 21, line 27: please replace "were" with "was".**

Done. Thank you for pointing this typo out.

---

## Author Comment (AC3) · 14 Feb 2019

**Responses to comments from Referee #3**

Thank you very much for your comments and suggestions. Providing this valuable feedback has helped to improve the current manuscript. We have modified the manuscript, taking into account all referee's suggestions. The following contains our detailed responses to your comments, with our responses in plain type given underneath your original comments in bold type.

**Overarching Comments**
**This paper describes a rigorous evaluation of the North et al. (2017) 3DVAR**
**multi-Doppler air motion retrieval using a high resolution cloud resolving model**
**and a radar-model simulator and carefully examining selected common sources of uncertainty:**
**characteristics of the scanning strategy, time updates of the scanning strategy, and advection**
**corrections to the data. The novel aspect of the paper is that the experiments can control for many**
**aspects of the sources of uncertainty given the high temporal resolution of the model output which**
**can be subsampled nearly arbitrarily for comparison with the retrievals. The paper makes the case**
**that for some characteristics of estimating vertical velocity in triple-Doppler networks, that having a**
**Doppler radar capable of scanning a PPI in less than 2 minutes is desirable, particularly for accurately**
**estimating the characteristics of weak updrafts.**

Thank you for the appreciation for this study.

**One major comment I have is that while the paper's methodology is thorough**
**and the figures are clear and illustrative, the presentation of the results is done in**
**an illogical fashion and does not allow for an easy read. The text refers to many**
**multi-panel figures in a repeated fashion that lead the reviewer to do a lot of page**
**flipping. I suggest breaking up the Figures 5, 6, 7, and 8 into different figures that**
**follow the logical flow of Section 3 without having to refer back to the figure subpanels over and over**
**again. Alternatively, discussion of the variables in Figs. 5-8 could be done sequentially by retrieved**
**variable rather than by observing strategy. I believe that the former strategy would be easier for the**
**authors to do.**
**I believe the paper is acceptable for publication with these minor revisions in**
**mind.**

Thank you for the suggestion. We broke Figures 6, 7, and 8 up into three figures:
Figure 6: UF, MF, w profiles for WRF outputs and 3FullGrid.
Figure 7: UF, MF, w profiles for 3XR and 3LR simulations and those for the limited area.
Figure 8: UF, MF, w profiles for 4SR simulations and 3XR simulations coupled with the advection correction.

**Specific Comments**
**1. Should the title be revised to include 3DVAR retrievals? The paper does not include other**
**traditional dual-Doppler retrievals such as in CEDRIC (which have**
**different assumptions about integrating the continuity equation than 3DVAR), or**
**more advanced analyses such as in SAMURAI (Bell et al.). Different techniques**
**could fare better or worse than the 3DVAR technique.**

Thank you for the suggestion. We changed the title to "Investigation of observational error sources in multi Doppler radar three-dimensional variational vertical air motion retrievals." Moreover, we referred to those papers (Miller and Fredrick, 1998 and Bell et al., 2012) in Introduction.

**2. It should be stated in the methodology that this analysis is based on a single case, and performance assessment could vary for different storm characteristics and propagation speeds. For example, a slow moving tropical squall line in low vertical wind shear could present less of a challenge for multi-Doppler retrievals than the highly-sheared May 20 MC3E squall line.**

Thank you for the valuable comment. We added a following sentence in Section 4: "The assessment of the multi-Doppler radar retrieval presented in this study could vary for different storm characteristics (e.g., isolated storm and less wind shear)."

**3. P5, L3, and elsewhere: the apostrophe is unnecessary.**

Done.

**4. P5, L25: suggest removing the word 'at'.**

Done.

**5. P6, L15-16: the CRSIM simulates the phase shift and the differential phase shift, not the phase and specific differential phase, correct?**

The CR-SIM can simulate the specific differential phase ($K_{DP}$) at each gridbox, but not the differential phase shift ($\phi_{DP}$).

**6. P7, L8: An analysis domain top of 10 km does not top the storm, correct? This would not allow for correct estimation of the storm-top divergence properly in the interpolated simulated radar data. Does this lead to some of the uncertainty unnecessarily?**

Correct; the top of the analysis domain (10 km) was not the storm top. Above 10 km altitude, the radar data density more decreases, and an uncertainty in the retrieved vertical velocity can increase with height, because these heights are poorly constrained by radial velocity observation. Collis et al. (2010) showed that radar mapping artifact where radar coverage is poor can lead to minimum vertical velocity errors of the order of 2 m/s at these heights. In our 3DVAR technique, the mass continuity constraint was applied at each grid box, and the calculation was performed until the cost function was minimized without such heights including poor radial velocity constraint. We tested a 3XRSnap simulation including higher altitudes upto 15 km. Figure below shows a comparison of the updraft fraction (UF) profiles with the WRF simulation (black lines) and the original 3XRSnap simulation (red lines). In the present case, updraft fractions for updrafts > 5 m/s and > 10 m/s above 10 km were smaller than 0.04 and 0.005, respectively. The updraft fractions for updrafts > 5 m/s from the new simulation (blue lines) converged

on the small updraft fractions above 10 km, unlike the profile from the original 3XRSnap simulation, However, at other altitudes, errors became larger compared to the original 3XRSnap simulation (e.g., 5-9 km and below 4 km for updrafts > 5 m/s, and 4-8 km for updrafts > 10 m/s).  We will need more analysis to address this impact in a separate paper. Thank you for the insight.

[Figure]

Fig. R2. Vertical profiles of updraft fractions with different thresholds of 5 m s$^{-1}$ (solid lines) and 10 m s$^{-1}$ (dashed lines). Black lines represent the WRF snapshot at 12:18:00 UTC, red lines represent the original 3FullGrid simulation presented in the manuscript, and blue lines represents the 3FullGrid simulation that the vertical domain was extended to 15 km AGL.

**7. P8, L20: "not unfolded correctly observed Doppler velocity" needs rewording.**

We revised the phrase to read "unfolding of observed Doppler velocity."

**8. P9, L14: "at" should be in.**

Done.

**9. P13, L2: The tilt would depend on the propagation speed and the vertical wind shear, which in some regimes (especially in the tropics) could be smaller than in this case.**

The referee is correct. We revised this sentence to read "horizontal advection and wind shear are expected to tilt the cloud and dynamical structures in vertical." Thank you for pointing this out.

**10. P13, L11: The smoothness function is a Cressman or Barnes filter? If not, what equation did you use?**

The advection correction procedure seeks to minimize a cost function that contains the frozen-turbulence constraint and so-called "penalty" terms that confer spatial smoothness on the pattern-translation components. The advection correction procedure is designed to produce smooth pattern-translation components U and V, but does not specifically attempt to smooth the scalar field being advected (i.e., reflectivity in the present study). The smoothness on the U and V components is obtained through the smoothness terms in the cost function, which are proportional to the squares of the horizontal gradients of U and V. As shown in Shapiro et al. (2010), the Euler-Lagrange equations arising

from minimization of this cost function contain terms (arising from the smoothness terms) that involve the Laplacians of U and V. Such terms act to "diffuse" the U and V fields, resulting in smooth U and V solutions. Theoretically, the procedure would preserve spatial discontinuities in the advected scalar field, if discontinuities were present in the input PPI data and if the scalar field satisfied the frozen-turbulence constraint.

For the referee's reference, we would like to show the cost function (Shapiro et al. 2010):

$$ J \equiv \iiint \left[ \alpha \left( \frac{\partial R}{\partial t} + U \frac{\partial R}{\partial x} + V \frac{\partial R}{\partial y} \right)^2 + \beta \left| \nabla_h U \right|^2 + \beta \left| \nabla_h V \right|^2 \right] dx \, dy \, dt $$

where U and V represent pattern-translation components, and R represents the advected scalar fields, which is the simulated radar reflectivity in the present study. The second and third terms are smoothness terms, and a coefficient β is the smoothness weighting coefficient.

In the revised manuscript, we added a following phrase to the first paragraph of section 2.4.4:

"The advection correction procedure seeks to minimize a cost function that contains the frozen turbulence constraint and terms that confer spatial smoothness on the pattern-translation components."

and revised the sentence to read:

"A weighting coefficient of the spatial smoothness terms in the cost function coefficient depends on the analysis grid spacing and the structure of the field being advected. An appropriate value of the coefficient can be determined by running some sensitivity tests. Based on preliminary tests (not shown), we deemed a coefficient of 300 dBZ^2 to be acceptable."

**11. P14, L1: Suggest inserting "potentially" before "large".**

We rephrased this as "large potential uncertainties." In the revised manuscript.

**12. P14, L25-26: "...but it tends to have higher uncertainty in the areas around the location of strong convection..." - this seems highly subjective. Can you make a quantitative statement related to this topic?**

The updraft fraction for 1-5 m/s from the 3XRSnap simulation was overestimated by $0.1 - 0.17$ above 2 km AGL, which accounts for 40-88% of that from the WRF output. The height of the error peak was around 5-6 km, which corresponds to the peak of $UF_5$ and $UF_{10}$. We added this description after this sentence.

**13. P15, L7: Regarding an updraft core defined as being larger than 0.5 km2: Is**

**this at a single level? This is one single pixel, correct? Since the retrieval has a resolution that is effectively 4-6 times the grid spacing, would one expect it to retrieve such small updraft features? Perhaps it is not surprising then that the retrievals have such issues resolving updrafts (particularly weak ones) that are single pixels in the grid? Perhaps increasing this threshold to have larger-sized updrafts (say larger than 2-3 km3) would reveal different uncertainties.**

This threshold corresponds to 8 pixels for the wind retrieval data and 2 pixels for the WRF data at a single level. The referee's comment is right; in the retrieval, a single data point could affect surrounding several grid points. Therefore we carefully decided this threshold to remove such noise in the retrieval results.

**14. P15, L16: Suggest inserting an "A" before "Noticable".**

Done.

**15. P16, L8-10: This sentence is awkward.**

We rephrased this sentence as "In subsequent sections, a more detail analysis of the impact of the different options in the observational setup on the UF, MF and $\overline{w}$ profiles are discussed."

**16. P20, L12-13: "cloud evolution cannot maintain the instantaneous cloud structures": Can you be more specific what you mean by this?**

We rephrased this sentence as "cloud evolution alters vertical and horizontal distributions of hydrometeors and vertical velocity, resulting in observing different cloud life stages by different PPI scans."

**17. P22, L6: "As previous literature has pointed out..."**

Done.

**18. P22, L8: 4.5 km is mid-levels, no?**

We changed it to "> 4.5 km."

**19. P22, L16: Need comma after "i.e."**

Done.

**20. P22, L18: "inferior" should be "lower"?**

The word "inferior" means "lower in quality" in this sentence. We would like to use this word here.

**21. P22, L28: Suggest replacing "hard" with "challenging".**

Done.

**22. P23, L10: Scott G.'s last name is misspelled.**

We are sorry about this. We corrected it in the revised manuscript.

**23. P23, L20: "DOW's" should be "Doppler on Wheels mobile radars"**

Done.

---

## Author Comment (AC4) · 14 Feb 2019

**Responses to comments from Referee #4**

Thank you very much for your comments and suggestions. We have modified the manuscript, taking into account all referee's suggestions. The following contains our detailed responses to your comments, with our responses in plain type given underneath your original comments in bold type.

**This paper investigates the impact of some sources of uncertainty on multiple-Doppler analysis of ground based radar data, with emphasis on vertical velocity. It is well written, clear and easily understandable. In fact, I don't have much to say about the points raised by the authors. The analysis process is good and the scientific argumentation is excellent.**
**Having said that, however, I am a little sceptical about the overall impact of this work and its general interest for the radar user community. Without being rude, it seems quite obvious that improving the VCP, the number of radars, or the time sampling resolution will have a positive impact on retrieved wind fields and, in particular, vertical velocity.**
**This has been shown previously by many authors in many different radar network configurations. While I agree that findings and recommendations resulting from this study would be useful for ARM RGP laboratory users, they would be difficult to apply to other networks. Furthermore using numerical model outputs and radar simulators to build a reference wind field is quite common nowadays and cannot be considered as a new concept.**
**Actually reading section 2.3 of the paper should generate quite some frustration among any scientist interested in radar wind retrieval. Indeed, questions regarding the impact of prescribed hydrometeor fall speed, potential masks, attenuation by rainfall (especially at X-band), or velocity folding count among the main sources of questioning for users and developers of multiple-Doppler analysis methods. To the best of my knowledge, these issues have never really been addressed to date and I was hoping that this study would help to clarify them, which would have contributed to make this paper a truly original contribution to the field. To be consistent with my remark, however, I must mention that the study on the impact of advection is original and does answer important questions.**

**To sum up, the work presented by the authors is of good quality, but its contribution to the field with respect to previous studies seems quite poor to me and results are barely applicable to networks other than ARM RGP laboratory. From there I see two options: 1/ Improving the paper by investigating additional sources of uncertainty such as Vt, velocity folding, attenuation... among others, or 2/ Clearly state that this study aims to optimize the performance of the ARM RGP network, without seeking to generalize its findings. Option 1 would require substantial additional work, but would undoubtedly represent an important contribution to the field. Option 2 would mostly imply cosmetic work (title, introduction, conclusions), but the impact of this paper would be limited.**

Thank you for the referee's appreciation for the study and pertinent comments. As the referee pointed it out, the analysis in this manuscript is limited to a 3DVAR multi Doppler radar technique and impacts of radar sampling limitation. First, we changed the title to "Investigation of observational error sources in multi Doppler radar three-dimensional variational vertical air motion retrievals."

We totally agree with the referee that the hydrometeor velocity estimates and the signal attenuation are major sources of errors in the wind retrieval. We have tested an impact of hydrometeor velocity estimate using reflectivity-based mass-fall velocity relationships proposed by Caya (2001) on the retrieved vertical velocity. Please see a response to the referee #2's second comment. The retrieved

vertical velocities from the simulation using the relationships were underestimated. However, it is hard to say whether the fall velocity estimate from these relationships or the hydrometeor fall velocity predicted by model simulation is more reliable. The present study focused on the uncertainties attributed to radar observation sampling, and we decided not to include the sensitivity test of the hydrometeor fall velocity estimate.

The (along-track) attenuation of hydrometeors can significantly impact the radar reflectivity measurements, but we confirmed that the reflectivity attenuation did not mask the Doppler velocity fields significantly in the analysis area. A figure attached below shows the simulated attenuated and non-attenuated radar reflectivity fields and associated Doppler velocities for the X-SAPR I4 radar at an elevation angle of 4.5 degrees. In all simulations in the present study, we used the Doppler velocity associated with the attenuated reflectivity. On the other hand, the attenuation in the reflectivity field can induce underestimations in the reflectivity-based hydrometeor fall velocity estimates. The underestimated hydrometeor fall velocity estimates could induce underestimation of the vertical velocity as shown in the response to the referee #2's second comment.

Our radar simulator does not include an option of simulating radial velocity folding, and we have not investigated an impact of the folded radial velocity on the vertical velocity retrieval. Recent velocity unfolding techniques can produce unfolded Doppler velocity fields with high accuracy (e.g., James and Houze, 2001). We expect that the folded Doppler velocity issue can be resolved in recent and future studies.

[Figure]

Fig. R3. PPI images of simulated reflectivity without attenuation (top left), Doppler velocity without reflectivity attenuation (top right), reflectivity with along-path two-way attenuation (bottom left), and Doppler velocity associated with the attenuated reflectivity (bottom left) for the X-SAPR I4 radar at an elevation angle of 4.5 degrees at 12:18:00 UTC.

As the referee pointed it out, the results of application of the advection correction to the wind retrieval are consistent with the previous studies; the advection correction is effective for shorter time period VCPs (< 2min). The present study also took into account the volume sampling and compared their impacts. The magnitude of improvement by the increase of elevation angles is larger than that by advection correction, even though the VCP needs 2 minutes. We modified the abstract and conclusion to more simplify them in response to other referee's comments and highlighted the result.

We also agree with the referee that the analysis in this manuscript is limited to the ARM radar network at the Southern Great Plains. Although the manuscript simulated the ARM precipitation radars, the similar radar network has been installed in many regions such as France (e.g., Bousquet et al. 2007), Germany (Helmert et al. 2014), and Japan (Maesaka, et al. 2011), and is expected to be extended to more countries. Moreover, future field campaigns targeting deep convection would be strongly motivated to install multiple Doppler radars to observe vertical air motions in convective clouds (e.g., https://www.arm.gov/news/features/post/52835). The present analysis can give valuable information to improve the observation strategies and decide optimized scan strategies for the networks. In the revised manuscript, we added a following sentence to the last paragraph: "Although the present study focused on the ARM X-band radar network, the similar dense radar network has been installed in several regions (e.g., Bousquet et al., 2007; Maesaka, et al. 2011; Helmert et al., 2014) and field campaigns targeting deep convection (past, on-going and future) would be strongly motivated to install multiple Doppler radars to observe vertical air motions in convective clouds. The present analysis can give valuable information to improve the observation strategies and decide optimized scan strategies for the networks."

From a cloud resolving model (CRM) perspective, the present study would notify the CRM communities of large uncertainties in the multi-Doppler radar-retrieved vertical air motion in upper parts of convective clouds, although some of the CRM simulation studies concluded that the CRMs significantly overestimated updrafts compared to multi-Doppler radar vertical velocity retrievals.

We have extended this OSSE study to a tropical convection case from Tropical Warm Pool – International Cloud Experiment and isolated storms over a Houston area. Those OSSEs commonly showed that limited radar sampling would cause underestimation of strong updraft areas. We are preparing separate papers for detailed analyses using the experiments.

**References:**
Bousquet, O., Tabary, P., and Parent du Châtelet, J.: On the value of operationally synthesized multiple-Doppler wind fields, Geophys. Res. Lett., 34, L22813, doi:10.1029/2007GL030464, 2007.
James, C.N. and Houze, R.A.: A Real-Time Four-Dimensional Doppler Dealiasing Scheme. J. Atmos. Oceanic Technol., 18, 1674–1683, https://doi.org/10.1175/1520-0426(2001)018<1674:ARTFDD>2.0.CO;2, 2001.
Helmert, K., and Coauthors: DWDs new radar network and post-processing algorithm chain. *Proc. Eighth European Conf. on Radar in Meteorology and Hydrology (ERAD 2014)*, Garmisch-Partenkirchen, Germany, DWD and DLR, 4.4, 2014. [Available online at http://www.pa.op.dlr.de/erad2014/programme/ExtendedAbstracts/237_Helmert.pdf]
Maesaka, T., Maki, M., Iwanami, K., Tsuchiya, S., Kieda, K., and Hoshi, A.: Operational rainfall estimation by X-band MP radar network in MLIT, Japan. Proc. 35th Int. Conf. on Radar Meteorology, Pittsburgh,

PA, Amer. Meteor. Soc., 142. 2011. [Available online at https://ams.confex.com/ams/35Radar/webprogram/Paper191685.html.]

**Other remarks:**
**1/ Results are based on a single case. Authors should keep that in mind in their conclusions. The effectiveness of vertical wind retrieval depends on many factors, including wind shear for example.**

Thank you for the important suggestion. Taking into account the referee #3's suggestion, we added a following sentence to the conclusion: "The assessment of the multi-Doppler radar retrieval presented in this study could vary for different storm characteristics (e.g., isolated storm and less wind shear)."

**2/ More details are needed regarding the wind retrieval method used in this paper. What about data interpolation and air-mass continuity equation (e.g., boundary conditions)?**

We used the 3DVAR multi-Doppler radar wind retrieval technique shown in North et al. (2017). This technique first needs to interpolate the radar data into the Cartesian coordinate. We used a Barnes distance-dependent weight (Eq. 1 in the revised manuscript) for the interpolation. The equation was applied in both horizontal and vertical interpolations. At each grid box, radar moments are estimated using the nearest 200 radar data gates with weights (Eq. 1) using a smoothing parameter $\kappa = 0.13$ km$^2$ for interpolation. The cutoff distance is determined as the distance where the weight is less than 0.01 (d $\approx 0.8$ km). These parameters are chosen so that the statistical error in retrieved vertical velocity is minimal for the present case. These settings for gridding are fixed for all radar simulations. We slightly modified Section 2.3 to briefly describe this gridding method.

As referee #1 pointed out, the gridding process can also be an error source in the vertical velocity retrieval. This error can be included and seen when we compare the 3FullGrid simulation with the other simulations using radar VCPs. We added this discussion to the second paragraph of Section 3.1 in the revised manuscript.

We used the 3DVAR wind retrieval technique described in North et al. (2017). The optimal wind field solution in the technique is obtained at the minimum of a cost function which contains the radial velocity observation constraint, anelastic mass continuity constraint, surface impermeability constraint, background wind field constraint, and spatial smoothness constraint. The mass continuity equation is an element of the cost function, and technique, the constraint was applied at each grid box. We used the surface impermeability constraint to dictate that vertical velocity vanishes at the ground with a relatively large weight. We modified Section 2.3 to read:
"The wind retrieval algorithm inputs the Cartesian coordinate $Z$ and $V_r$ fields from each radar and uses 3DVAR technique continuity constraint proposed by Potvin et al. (2012a). In the technique, the optimal wind field solution in the technique is obtained at the minimum of a cost function which consists of the physical constraints of radar radial velocity observations, anelastic mass continuity, surface impermeability, background wind field, and spatial smoothness. The surface impermeability constraint was used to dictate that vertical velocity vanishes at the ground with a relatively large weight."

**3/ More details are needed about the investigated weather system. Authors should include additional material to better describes the overall structure of the storm (e.g. vertical cross-section of model/radar reflectivity/wind fields).**

We added short descriptions about the observed MCS and WRF-simulated MCS to the first paragraph of Section 2 and Section 2.1, respectively.

For the observed MCS, we added: "This squall-line MCS was oriented in northeast-southwest direction extending for approximately 1000 km (Fan et al., 2017). The convective region had approximately 50 km width and trailed a distinct stratiform precipitation area when it passed through the ARM SGP site from 09:20 UTC to 11:40 UTC."

For the WRF-simulated MCS, we added: "The simulated MCS comprised a convective precipitation region at the leading edge of the system and a stratiform precipitation trailed by the convective region, as similar as the observation. The MCS passed through the ARM SGP radar observation site approximately one hour later than the observation (at around 12:18 UTC), and a stronger convective precipitation region formed slightly (~20 km) to the north of the ARM SGP site."

Moreover, we referred previous studies by Liu et al. (2015), Wu and McFarquhar (2016), and Fan et al. (2017), where dynamical and microphysical structures were analyzed.

---

## Author Comment (AC5) · 14 Feb 2019

Attached is the revised manuscript.

Please also note the supplement to this comment:
https://www.atmos-meas-tech-discuss.net/amt-2018-442/amt-2018-442-AC5-supplement.pdf

———————————————————

---

## Author Comment (AC6) · 14 Feb 2019

[revised manuscript text omitted]

---

## Author Comment (AC7) · 14 Feb 2019

[revised manuscript text omitted]

MF5 [kg m-2 s-1]  $MF_{10}$  [kg m-2 s-1]  $\overline{W}_{10}$  [m s-1]  $\overline{W}_{5}$  [m s-1] UF5  $UF_{10}$  $0.34 \times 10^{-2}$  $0.18 \times 10^{-2}$  $0.23 \times 10^{-1}$  $0.18 \times 10^{-1}$ 3FullGrid 0.15 1.93 3.06×10-2 0.68×10-2 0.89×10-1 0.60×10-1 3XRSnap 1.18 0.65 3.45×10-2 0.63×10-2 1.02 ×10-1 0.52×10-1 1.10 3XR2min 0.53 3XR5min 3.26×10-2 1.03×10-2 1.08 ×10-1 1.48 1.01  $0.95 \times 10^{-1}$ 1.99×10-2 0.42×10-2 0.68 ×10-1 0.37×10-1 0.50 3LRSnap 0.87 3LR2min  $2.44 \times 10^{-2}$ 0.49 ×10-2  $0.85 \times 10^{-1}$ 0.41×10-1 0.91 0.52 3LR5min 3.94×10-2 0.92×10-2 1.37 ×10-1 0.75×10-1 1.23 0.70 0.64×10-2 3.57×10-2 1.04×10-1 0.56×10-1 4SRSnap 1.12 0.67 1.04 ×10-1 4SR2min 3.43×10-2 0.66×10-2 0.57×10-1 1.06 0.56 4SR5min 1.10×10-2 1.91×10-1 0.83×10-1 5.79×10-2 1.33 0.81 0.75×10-2 3XR2minadv 2.90×10-2 0.96×10-1 0.65×10-1 1.11 0.61 1.10×10-1 3.38×10-2 0.96×10-2 0.88×10-1 1.28 0.89 3XR5minadv 3LR2minadv 1.39×10-2 0.71×10-2 0.64×10-1 0.64×10-1 0.90 0.66 1.55×10-2 1.15×10-2 0.85×10-1 1.02×10-1 1.40 0.85 3LR5minadv 3XRSnap 5.09×10-2 1.29×10-2 1.53×10-1 1.08×10-1 1.06 2.86 (limited area) 4.73×10-2 1.48×10-2 3XR2min 1.69×10-1 1.20×10-1 0.86 0.96 (limited area) 2.24×10-2 0.79×10-1 3LRSnap 0.89×10-2 0.83×10-1 0.71 2.87 (limited area) 2.19×10-2 0.84×10-1 1.28×10-2 1.19×10-1 0.81 3LR2min 0.88 (limited area) 5.83×10-2 1.11×10-2 1.74×10-1 4SRSnap 0.94×10-1 0.93 2.84 (limited area)

Table 3: Root mean square error (RMSE) of UF5, UF10, MF5, MF10,  $\overline{w}_5$ , and  $\overline{w}_{10}$  profiles above 2 km AGL for all experiments.

---

## Author Comment (AC9) · 17 Feb 2019

Thank you for your valuable comments and suggestions to improve our manuscript.